# Investigating the etiologies of non-malarial febrile illness in Senegal using metagenomic sequencing

Zoë C. Levine [1,2,3], Aita Sene[4,5], Winnie Mkandawire[1,6], Awa B. Deme[5], Tolla Ndiaye[4,5], Mouhamad Sy[4,5], Amy Gaye[4,5], Younouss Diedhiou[4,5], Amadou M. Mbaye[4,5], Ibrahima M. Ndiaye[4,5], Jules Gomis[4,5], Médoune Ndiop [7], Doudou Sene[7], Marietou Faye Paye[1], Bronwyn L. MacInnis [1,8], Stephen F. Schaffner [1,8,9], Daniel J. Park [1], Aida S. Badiane[4,5], Andres Colubri[1,6], Mouhamadou Ndiaye[4,5], Ngayo Sy[10], Pardis C. Sabeti [1,8,9,11,13] ✉, Daouda Ndiaye[4,5,13] ✉ & Katherine J. Siddle [1,12,13] ✉

The worldwide decline in malaria incidence is revealing the extensive burden of non-malarial febrile illness (NMFI), which remains poorly understood and difficult to diagnose. To characterize NMFI in Senegal, we collected venous blood and clinical metadata in a cross-sectional study of febrile patients and healthy controls in a low malaria burden area. Using 16S and untargeted sequencing, we detected viral, bacterial, or eukaryotic pathogens in 23% (38/163) of NMFI cases. Bacteria were the most common, with relapsing fever *Borrelia* and spotted fever *Rickettsia* found in 15.5% and 3.8% of cases, respectively. Four viral pathogens were found in a total of 7 febrile cases (3.5%). Sequencing also detected undiagnosed *Plasmodium*, including one putative *P. ovale* infection. We developed a logistic regression model that can distinguish *Borrelia* from NMFIs with similar presentation based on symptoms and vital signs (F1 score: 0.823). These results highlight the challenge and importance of improved diagnostics, especially for *Borrelia*, to support diagnosis and surveillance.

Febrile illness is a significant cause of morbidity and mortality in West Africa. While malaria remains the most common single pathogen causing febrile illness, its incidence has decreased sharply in the last two decades; in Senegal, for example, control measures decreased malaria incidence from 122 per 1000 in 2006 to 59 per 1000 in 2021[1,2]. With declining incidence and increased use of malaria rapid diagnostic tests, the importance of non-malarial febrile illness (NMFI) has become

more apparent. Unlike malaria, however, many of the pathogens causing NMFI are not the target of robust surveillance programs, and rapid diagnostic tests (RDTs), the backbone of both clinical care and surveillance in peripheral care settings, are not available[3].

In the absence of comprehensive surveillance efforts that capture multiple pathogen types, appropriate public health interventions are hindered by our limited understanding of the full landscape of

[1]Broad Institute of Harvard and MIT, Cambridge, MA, USA. [2]Harvard Graduate Program in Biological and Biomedical Science, Boston, MA, USA. [3]Harvard/MIT MD-PhD Program, Boston, MA, USA. [4]Department of Parasitology, Cheikh Anta Diop University Dakar, Dakar, Senegal. [5]Centre International de Recherche et de Formation en Génomique Appliquée et de la Surveillance Sanitaire, Dakar, Senegal. [6]University of Massachusetts Medical School, Worcester, MA, USA. [7]Programme National de lutte contre le Paludisme, Ministère de la Santé, Dakar Fann, Senegal. [8]Department of Immunology and Infectious Diseases, Harvard T.H. Chan School of Public Health, Harvard University, Boston, MA, USA. [9]Department of Organismic and Evolutionary Biology, Harvard University, Cambridge, MA, USA. [10]Service de Lutte Anti Parasitaire, Thies, Senegal. [11]Howard Hughes Medical Institute, Chevy Chase, MD, USA. [12]Department of Molecular Microbiology and Immunology, Brown University, Providence, RI, USA. [13]These authors jointly supervised this work: Pardis C. Sabeti, Daouda Ndiaye, Katherine J. Siddle. ✉e-mail: pardis@broadinstitute.org; daouda.ndiaye@cigass.org; katherine_siddle@brown.edu

common causes of NMFI at the community level. Untargeted sequencing, also known as metagenomic sequencing (mNGS), is a powerful tool for detection of microbial nucleic acids in clinical samples without a priori knowledge of a pathogen[4,5] and is increasingly used for surveillance in regions at high risk of emerging and reemerging disease[6–8]. As mNGS sequences all RNA or DNA in a sample, these techniques are typically less sensitive for detecting any single pathogen than targeted approaches (e.g., PCR), due to the abundance of the host relative to pathogen nucleic acids[4,5]. In order to achieve higher sensitivity and reduce the cost of deep sequencing, mNGS can be applied to cell-free fluids, which have lower host backgrounds, but this can limit the detection of intracellular pathogens. Untargeted approaches are complemented by more targeted strategies, such as 16S sequencing for bacterial pathogens, that specifically amplify pathogen nucleic acids, reducing required sequencing depth and simplifying interpretation. This technique has been applied to detect bacterial bloodstream infections[9,10] and tick-borne bacterial illness[11].

A broad range of pathogens across kingdoms can cause NMFI, but surveillance studies often focus on one or a few pathogens or are limited to hospitalized patients and, therefore, are limited in their ability to detect all causes of disease. Surveillance studies of severely ill hospitalized patients have identified bacterial pathogens, including several vaccine-preventable illnesses such as *Streptococcus pneumoniae* and *Neisseria meningitidis*[12,13]. Arboviruses, such as Dengue virus and Chikungunya virus have been detected in time-limited outbreaks in Senegal[14,15]. Community and clinic-based studies of patients with fever have revealed bacterial zoonoses are a common cause of ambulatory febrile illness in Senegal, including tick-borne relapsing fever, Rickettsioses, Q fever, and *Bartonella*[16–21]. However, no unbiased surveillance of outpatient febrile illness has been done in Senegal to date.

This study aimed to determine and characterize causes of ambulatory NMFI in Thiès, Senegal, a peri-urban community with overall low malaria incidence where malaria transmission dynamics have been deeply characterized[22]. We collected plasma samples from a population of acutely febrile patients presenting to the Service de Lutte Anti Parasitaire (SLAP) outpatient clinic and healthy controls from the surrounding neighborhoods during the dry and rainy seasons of 2018 and 2019. We aimed to provide insights on clinical presentations of NMFI that can guide providers at the point of care and genomic characterization to inform the design of new detection tools for clinical diagnosis and public health surveillance.

## Results

### Characterization of NMFI reveals viral, bacterial, and eukaryotic pathogens

We first characterized the pathogens in plasma samples collected from acutely febrile patients suspected of malaria and healthy controls across the dry (febrile: $n = 100$, healthy: $n = 54$) and rainy (febrile: $n = 104$, healthy: $n = 50$) seasons in 2019 using both untargeted and targeted approaches (Supplementary Fig. 1a, Supplementary Fig. 1b). All febrile patients received a malaria RDT; 39.4% (41/104) of patients tested positive for malaria in the wet season and no cases were detected in the dry season. Febrile cases were roughly equally split between children and adults (54% ≥18 yrs). Age distribution was similar ($p = 0.98$, Mann–Whitney, two-sided) across case and control groups, but there were more male febrile cases (Male: $n = 111$, Female: $n = 93$) and more female controls (Male: $n = 47$, Female: $n = 57$, $p = 0.15$ Fisher exact two-sided, Supplementary Fig. 1a).

We detected viral, bacterial, and eukaryotic pathogens by sequencing. To detect viral pathogens, we performed RNA-mNGS. We also evaluated the ability of RNA-mNGS to detect non-viral pathogens, including malaria and fungi. To detect bacterial infections, we sequenced the v1–2 region of the 16S rRNA gene, which permitted us to classify the bacterial taxa present in samples with high bacterial load [see methods]. We detected at least one pathogen in the plasma in 23%

(38/163) of RDT-negative acutely febrile patients and co-infections in 7% (3/41) of RDT-positive acutely febrile patients (Fig. 1b). The most common febrile pathogen was *Borrelia*, which was found across the dry ($n = 8$) and rainy ($n = 10$) seasons and in both RDT-negative ($n = 17$) and RDT-positive ($n = 1$) patients. We also detected bacterial infections with *Rickettsia* and *Arcobacter* (Fig. 1d). Bacterial pathogens found as co-infections included two *Borrelia/Rickettsia* co-infections, two *Borrelia/Plasmodium* co-infections, and three *Plasmodium/Rickettsia* co-infections (Fig. 1e). We did not detect any viral/bacterial or viral/ *Plasmodium* coinfections.

We identified four known vertebrate viruses: Dengue virus (DENV, $n = 2$), Hepatitis B virus (HBV, $n = 2$), Parvovirus B19 ($n = 2$), and Human immunodeficiency virus 1 (HIV-1, $n = 1$) as well as a human virus not currently believed to cause disease, Human pegivirus 1 (HPgV-1, $n = 4$) (Fig. 1c). Four febrile patients and one healthy control exhibited a high proportion of fungal RNA reads in the plasma, as compared to healthy controls (>99th percentile reads/ million raw reads (rpm) for healthy controls), but these reads did not map to any specific fungal pathogen (Supplementary Fig. 2). We did not detect any vertebrate viruses in healthy controls, but we did detect *Borrelia* ($n = 1/35$ samples sequenced) and *Rickettsia* ($n = 1/ 35$) (Supplementary Table 1).

### RNA-mNGS identifies viral pathogens known to circulate in Senegal

We next considered the genetic diversity of detected viral pathogens to determine their relationship to other circulating strains in West Africa. Phylogenetic analysis of complete DENV, HBV, and Parvovirus B-19 genomes indicated that in all three cases, the two genomes were different genotypes and thus not closely related (Supplementary Table 2). Notably, there was a DENV outbreak across Senegal, including Thiès, in 2018. Whole genome phylogenetic analysis revealed the closest relatives to the 2019 DENV3 genome from this study were DENV3 genomes from patients presenting to the SLAP clinic during that outbreak[14] (Supplementary Fig. 3a). Conversely, the DENV1 genome in this study was more closely related to other genomes from West Africa than to the DENV1 genomes from the 2018 outbreak (Supplementary Fig. 3a). The HIV-1 genome could not be assembled or genotyped due to low read count (mean coverage 0.2), but reads mapped across the reference genome including in the *gag, pol*, and *env* genes.

While HPgV-1 is known to infect asymptomatic individuals across the world, we only detected HPgV-1 in febrile patients. The prevalence of HPgV-1 has not been well studied in Senegal; we found infection in only 1.3% (4/307) of plasma samples sequenced, lower than the 4–11% reported in blood-donor surveys from Sub-Saharan Africa[23,24]. All four genomes were clustered with publicly available genomes from human hosts in Sierra Leone, Uganda, and Cameroon and belonged to Genotype I (Supplementary Fig. 3b), the most commonly circulating HPgV-1 genotype in West Africa[25].

In order to identify divergent viral species missed at the read level, we performed a translated nucleic acid search of de novo contigs, which identified RNA-dependent RNA Polymerase (RdRp) sequences for two candidate novel viruses, one in the *Naranviridae* family and one in the *Reoviridae* family. We detected a *Reoviridae* RdRp sequence across 18 febrile patients, 12 healthy controls, and 4 non-template controls, suggesting a likely contaminant (Supplementary Fig. 4). However, given that *Reoviruses* have been isolated from ill patients[26], we further investigated these sequences. The *Reovirus* RdRp sequences from this study clustered together but were distant from mammalian *Orthoreoviruses*. Given the distance from mammalian *Orthoreovirus* species and presence in non-template controls, these sequences were likely from a contaminated reagent rather than true human infections. We also identified a *Narnaviridae* RdRp sequence across 5 febrile patients

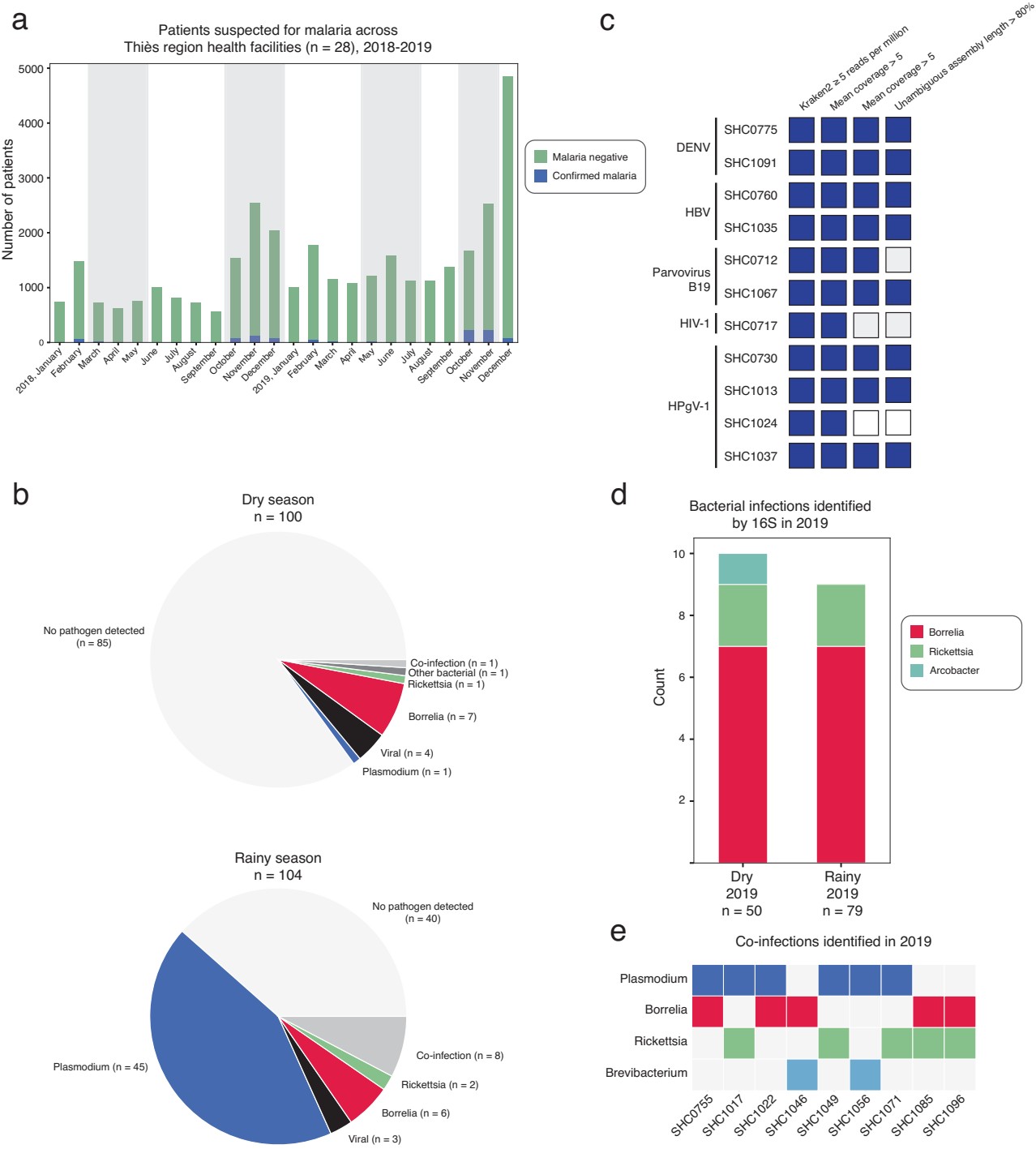

**Fig. 1 | Overview of the burden of malaria and NMFI in Thiès, Senegal and NMFI pathogens identified in 2019. a** Patients suspected of malaria across 28 health facilities in the Thiès region (data provided by Program National de lutte contre le Paludisme); gray shading indicates periods of sample collection for this study. **b** Types of pathogens detected in febrile patients in dry and rainy seasons in 2019. *Plasmodium* detected by RDT or RNA-mNGS, bacterial infections detected by 16S amplicon sequencing, and viral infections detected by RNA-mNGS. **c** Viral infections detected by RNA-mNGS. **d** Bacterial infections detected by 16S sequencing and **e** Co-infections detected in febrile patients from 2019. Source data are provided as a Source Data file.

and 1 healthy control (Supplementary Fig. 5), none of whom had any other identifiable pathogen. Because viruses in the *Narnaviridae* family, which infect plants and fungi, have not been reported as human pathogens and we detected this novel *Narnavirus* species in both febrile cases and a healthy control, the role of this virus in disease, if any, is unclear.

**Untargeted RNA sequencing detects *Plasmodium* cases missed by RDT**

If RNA-mNGS is to be employed as a tool for the diagnosis of febrile illness in malaria-endemic regions, we also need to understand the ability of RNA-mNGS to detect malaria as compared to clinical diagnostics. We know that RDTs have limited sensitivity, detect only certain

*Plasmodium* species, and are susceptible to false negatives in genetically diverse parasites, so we evaluated our RNA-mNGS reads for evidence of *Plasmodium* infections.

RNA-mNGS detected abundant *Plasmodium* nucleic acids (>550 rpm) in 6.1% (10/163) of RDT-negative acutely febrile patients and one healthy control (1/104), suggesting these were false negatives. Among RDT positive patients, there was a wide range in the abundance of *Plasmodium* reads (mean $1.7 \times 10^4$ rpm, range $1.5 \times 10^2 - 2.6 \times 10^5$ rpm) (Supplementary Fig. 6a). Overall, 63% of RDT positive patients (26/41) and 77% of smear-positive patients (17/22) were also positive by untargeted RNA sequencing (>550 rpm) (Supplementary Fig. 6c). In patients with detectable parasitemia by blood smear examination, *Plasmodium* reads did not correlate strongly with parasite density (Pearson's $R = 0.49$, Supplementary Fig. 6b).

In order to determine why 11 samples were negative by RDT but positive by RNA-mNGS, we first confirmed the results with a pan-*Plasmodium* qPCR. We found that 8/11 samples were positive by pan-*Plasmodium* qPCR, and the cycle threshold (CT) correlated with *Plasmodium* RNA-mNGS rpm (Supplementary Fig. 6d). Next, we determined whether any samples belonged to species other than *Plasmodium falciparum*, the target species of the RDTs, using both RNA-mNGS reads and a *P. falciparum*-specific qPCR. As previous work has demonstrated, Kraken2 is inaccurate for species-level classification of parasites[27]; we used DIAMOND to identify one sample with a contig (771 bp) that matched perfectly (100% nucleotide identity, 100% coverage) to *Plasmodium ovale* cytochrome oxidase subunit 1 (Accession: KP050416.1), suggesting a likely *P. ovale* infection. This sample was positive on the pan-*Plasmodium* qPCR assay but negative on the *P. falciparum* qPCR, confirming it as a case of non-*falciparum* malaria (Supplementary Table 3, Supplementary Fig. 6e).

Additionally, we assessed expression levels of histidine-rich protein 2 (PfHRP-2), the target antigen of the RDTs used in this study. Deletions in *pfhrp2* have been detected in Senegal and demonstrated to cause decreased antigen expression and consequently false negative RDT results[28,29]. The number of RNA-mNGS rpm aligned to *pfhrp2* was significantly different between patients who were RDT positive and RNA-mNGS positive (RDT+/mNGS+) and patients who were RDT negative but *Plasmodium* RNA-mNGS positive (RDT-/mNGS+) (Supplementary Fig. 6f, Mann Whitney two-sided, $p = 0.038$). To assess for the presence of deletions at common breakpoints, we amplified two targets, one in exon 2 of *pfhrp2* gene and one in a flanking gene, which can also be included in deletions[30]. Amplification of both targets was intact in all qPCR-confirmed *P. falciparum* infections (7/7, Supplementary Fig. 6g), suggesting none of these parasites had deletions in these targets consistent with the low frequency of *pfhrp2* deletion in Senegal, previously estimated at 2.4%[29].

## 16S sequencing confirms a high burden of relapsing fever *Borrelia* and spotted fever *Rickettsia*

As bacteria were the most common cause of NMFI in our 2019 study population, we extended our 16S sequencing to detect bacterial infections across the complete study population from 2018 to 2019 and further investigated the species causing the disease. Among all high bacterial load samples that underwent 16S sequencing [see methods], we detected *Borrelia* in 15.5% (33/213) and *Rickettsia* in 3.8% (8/213) of febrile patients (Fig. 1d, Supplementary Fig. 7c). 16S sequencing also identified other bacterial genera containing known pathogens, including *Brevibacterium*, *Arcobacter*, and *Veillonella*, across both years (Supplementary Fig. 7c). However, given the limited taxonomic resolution of v1-2 16S sequencing, we were unable to determine whether these sequences represented pathogenic species or harmless commensals.

Phylogenetic clustering of *Borrelia* v1-2 16S sequences showed that all *Borrelia* sequences from this study were similar to each other and fell within the relapsing fever (RF) group, and the majority were

most similar to *B. crocidurae* (Supplementary Fig. 8a). RF *Borrelia* circulate worldwide and are known to cause febrile illness in Senegal[16-20]. The *Rickettsial* v1-2 16S rRNA gene sequences from our study clustered within the Spotted Fever group, most closely related to *R. felis* (Supplementary Fig. 8c). Spotted fever *Rickettsia*, including *R. felis*[21] and *R. africae*[17] have been previously detected in febrile Senegalese patients.

To validate the 16S v1-2 phylogeny, a subset of 6 *Borrelia* positive samples were typed by amplicon sequencing of the 16S-23S intergenic spacer (IGS)[31]. All sample sequences were *Borrelia crocidurae*, in agreement with the 16S v1-2 phylogeny (Fig. 2c, Supplementary Fig. 8b). Sequences from this study were more similar to published sequences from humans and *Ornithodoros sonrai* in Southern Senegal (unpublished), and *Ornithodoros sonrai* ticks in Mali[32] than sequences from *Ornithodoros erraticus* in Tunisia[33] and Morocco[34] (Fig. 2c).

## Sensitive detection of *Borrelia* by 16S and RNA-mNGS

We assessed the ability of RNA-mNGS to detect bacterial infections based on samples that underwent both sequencing methods. *Borrelia* RNA-mNGS reads were detected in 68% (13/19) of 16S positive samples and 100% (17/17) of pan-*Borrelia* qPCR positive samples (Supplementary Fig. 7i). Conversely, *Rickettsia* RNA-mNGS reads were not detected in any *Rickettsia* positive samples. Acute RF *Borrelia* infection is known to cause high titers of bacteria in the blood during febrile episodes, while *Rickettsia* is an obligate intracellular pathogen and, therefore, has low titers in the cell-free plasma[35]. To assess the extent to which plasma titers impacted the sensitivity of RNA-mNGS to tick-borne bacterial pathogens, we quantified bacterial abundance by the percent of total v1-2 16S sequences from a given sample classified as *Borrelia* or *Rickettsia*, respectively[11]. We observed *Borrelia* abundances ranging from 5% to 98%, while *Rickettsia* abundances were lower, ranging from 5% to 16%. RNA-mNGS only detected *Borrelia* in samples with abundance >20% (Supplementary Fig. 7f). Taken together, these data suggest that plasma RNA-mNGS can detect high titer bacterial pathogens but will miss less abundant intracellular species.

We compared the sensitivity of our 16S sequencing to qPCR and Giemsa stained blood smear examination, the current gold standard for clinical diagnosis of RF *Borrelia*[36]. We examined smears from a subset of RF *Borrelia* patients. 28.6% (4/14) of examined smears were positive; all 4 positive patients had a high RF *Borrelia* load (≥90%, Mann-Whitney two-sided $p = 0.002$ compared to 16S positive/smear negative) (Supplementary Fig. 7b, d). Although blood smear examination is the most widely used diagnostic, qPCR diagnostics are known to have higher sensitivity[36,37]. Using a pan-*Borrelia* qPCR assay previously used for the detection of *Borrelia* in patient blood samples[38], 79% (27/34) of *Borrelia* samples were confirmed by qPCR, while six samples that were negative by 16S sequencing were positive by qPCR. Bacterial load measured by qPCR and 16S abundance correlated well (Pearson's $R = 0.72$, Supplementary Fig. 7g). While 16S sequencing was done only on a subset of samples, pan-*Borrelia* qPCR was performed on all febrile cases and healthy controls in 2019 and all febrile cases in 2018. An additional five cases of *Borrelia* were identified in febrile patients by qPCR in samples that did not undergo 16S sequencing.

## *Borrelia* infection presents similarly to other febrile illnesses but can be distinguished by key features

We sought to assess the clinical syndrome associated with qPCR-confirmed *Borrelia* infections, compared to RDT-confirmed malaria and non-*Borrelia* NMFI ("other febrile"), to guide differential diagnosis at the point of care. *Borrelia* infections occurred in a consistent proportion of the study population in the two years (Fisher exact $p = 0.61$) and in both seasons (Fisher exact $p = 0.24$). *Borrelia* infections occurred across all ages (*t*-test compared to all other febrile $p = 0.45$) (Fig. 2b) and at similar rates in male and female patients (Fisher exact

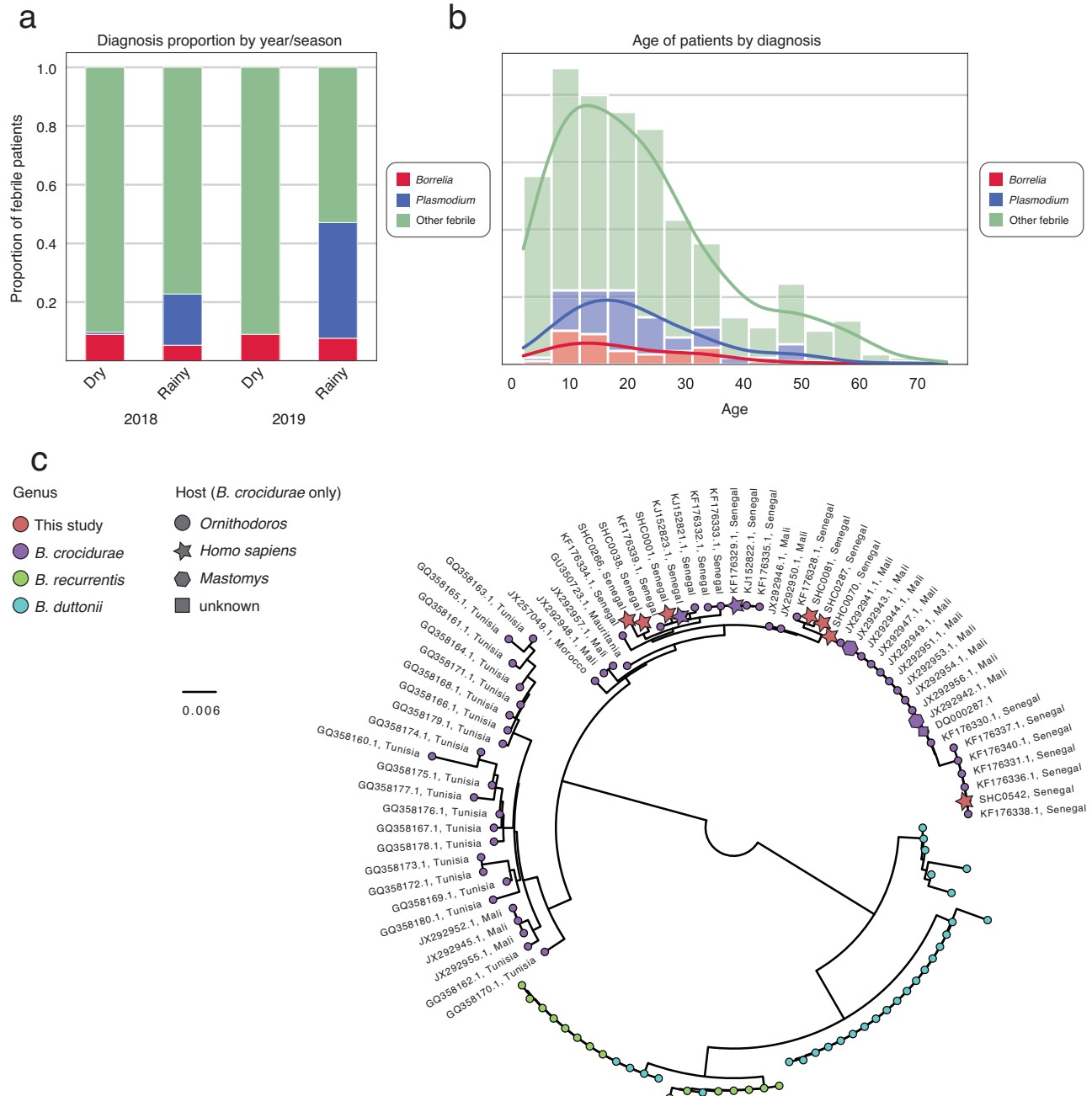

**Fig. 2 | Characterization of qPCR-confirmed *Borrelia* and RDT-confirmed *Plasmodium* across both years. a** Proportion of the study population testing positive for *Borrelia*, *Plasmodium*, or neither (Other febrile) across dry and rainy seasons in 2018 and 2019. **b** Age distribution of *Borrelia*, *Plasmodium*, and other febrile patients. **c** Maximum likelihood phylogenetic tree (midpoint rooted) of IGS sequences from this study (red) in the context of available reference sequences for *B. crocidurae* (purple), *B. duttonii* (cyan), and *B. recurrentis* (green). For *B. crocidurae* isolates, the location of collection is indicated in the label, and host species are indicated with node shape as available. Source data are provided as a Source Data file.

$p = 0.87$), while male patients were more likely to test positive for malaria (Fisher exact $p = 1.6e - 7$).

A high proportion of *Borrelia*-positive patients reported generalized symptoms that were common across all febrile patients, including headache (100%), body aches (59%), and dizziness (46%), as reported in previous studies of RF *Borrelia* (Fig. 3a)[36]. *Borrelia* infection was distinguished by vomiting (49%), which was observed more often in *Borrelia*-positive patients (stat = 2.8, $p = 0.0026$, Fisher Exact two-sided). *Borrelia*-positive patients were less likely than other febrile patients to report sore throats (7.7%, stat = 0.21, $p = 0.0039$, Fisher Exact two-sided). Notably, RF Borrelia can invade the central nervous

system in severe cases; while we did not observe neurological symptoms such as seizures or loss of consciousness, one *Borrelia*-positive patient reported eye pain, a potential symptom of neuroinvasive infection not observed in any other febrile patients[36,39]. A high proportion of *Borrelia*-positive patients reported contact with a febrile person (38%), contact with rats (57%), and prior travel (32%), but these exposures were common across malaria and other febrile patients and were not associated with *Borrelia* infection in particular (Fig. 3b).

*Borrelia* patients showed significant differences in temperature and blood glucose. On average, *Borrelia*-positive patients had a higher fever (mean = 38.6 °C, SD = 1.1 °C) than other febrile patients (mean =

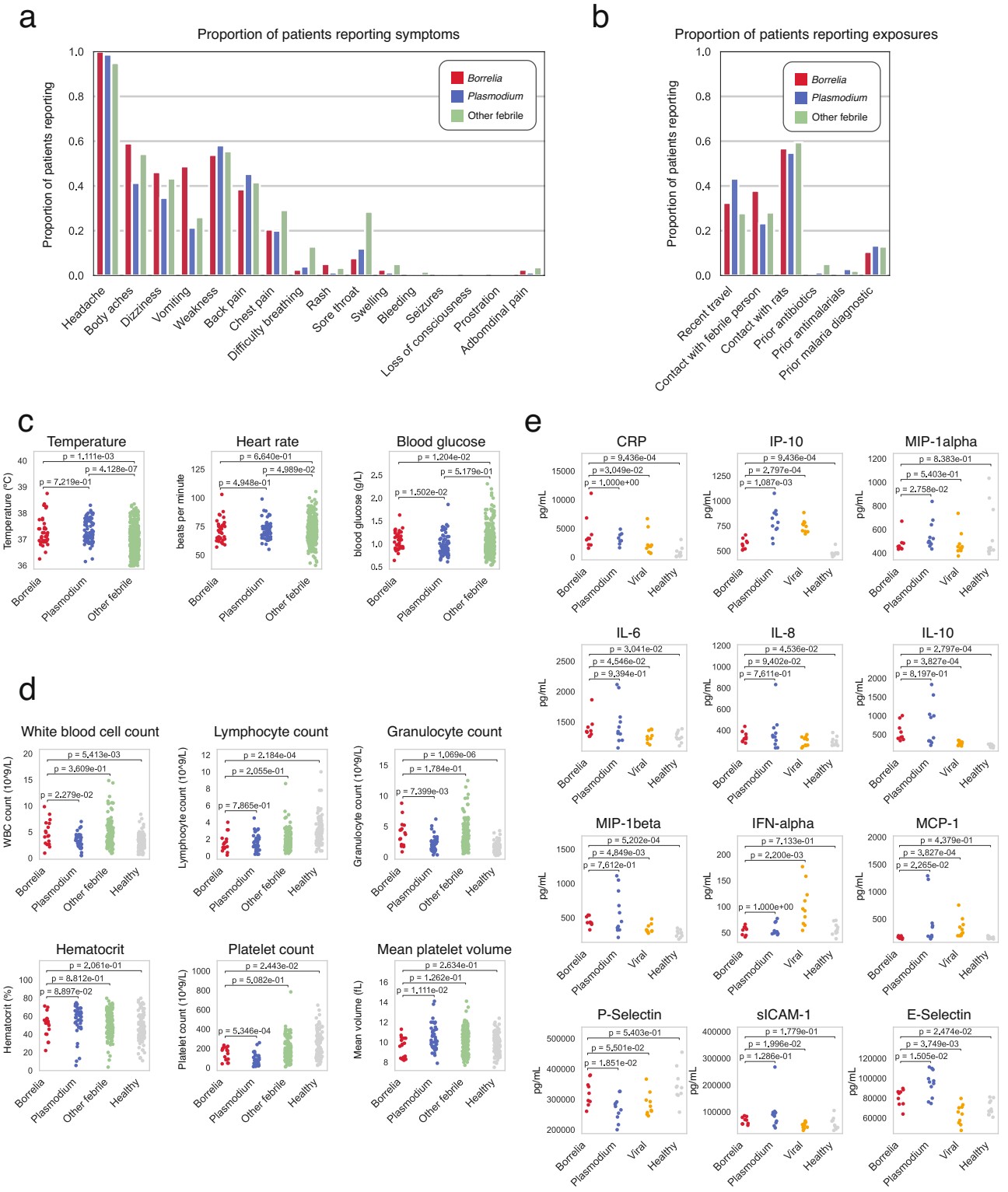

**Fig. 3 | Signs, symptoms, and exposures in *Borrelia* positive, *Plasmodium* positive, and other febrile patients.** Clinical characteristics across both years of *Borrelia* (n = 39), *Plasmodium* (n = 75), and other febrile (n = 412) patients, including **a** symptoms reported by patients, **b** exposures reported by patients, and **c** vital signs. For 2019 only, **d** blood cell counts in *Borrelia* (n = 17), *Plasmodium* (n = 41), other febrile (n = 146), and healthy (n = 104). **e** Serological profile of a subset of patients with *Borrelia* (n = 9), *Plasmodium* (n = 11), viral infection (n = 10) compared to healthy patients (n = 10); cytokines/chemokines with at least one significant difference between groups shown [see Source Data for results for all cytokines/ chemokines tested]. All *p*-values represent the result of the Mann–Whitney–Wilcoxon two-sided test. Source data are provided as a Source Data file.

38.0 °C, SD = 1.0 °C, Mann–Whitney two-sided p = 0.0011). *Borrelia* patients (mean = 1.07 g/L, sd = 0.19) were also hypoglycemic compared to other febrile patients (Other febrile: mean = 1.22 g/L, sd = 3.97, Mann–Whitney two-sided p = 1.204e − 02) (Fig. 3c).

We assessed whether the addition of a complete blood cell count with differential could help distinguish RF *Borrelia* from other febrile diseases. *Borrelia*-positive patients exhibited abnormal blood counts, including lymphopenia (mean $1.5 \times 10^9$ cells/L, std 1.04),

granulocytosis (mean $7.47 \times 10^9$ cells/L, std 4.23), and thrombocytopenia ($148 \times 10^9$ cells/L, std 66) (Fig. 3d). However, this hematologic response was not significantly different from the response observed in other febrile patients. On a subset of patients, we measured 20 cytokines and chemokines to further distinguish *Borrelia* from other infections (Fig. 3e). *Borrelia*-positive patients were distinct from viral infections in elevated CRP (Mann–Whitney two-sided, $p = 0.030$), IL-10 ($p = 0.00038$), and MIP-1Beta ($p = 0.0048$) and decreased IP-10 ($p = 0.00028$), IFN-alpha ($p = 0.0022$), and MCP-1 ($p = 0.00038$) (Fig. 3e).

Given the highly overlapping presentation of the disease, we developed a weighted logistic regression model to distinguish *Borrelia* infection from other NMFI based on clinical symptoms, vital signs, and key demographic information. We evaluated the performance of the model with bootstrapping and found the model was able to predict *Borrelia* infection with high performance (recall: 0.861, 95% CI [0.837–0.910], precision: 0.792, 95% CI [0.696–0.892], and F1-score: 0.823, 95% CI [0.766–0.900]) (Fig. 4a). The model identified features that were significant in univariate analysis, including vomiting, temperature, blood glucose, and sore throat. Common symptoms that were observed across all febrile patients, such as headache, body ache, and dizziness, were not useful predictors on their own but were associated with an increased risk of *Borrelia* when evaluated in the context of other symptoms and exposures. To test whether additional laboratory tests might further increase the performance of the model, we incorporated complete blood count with differential (CBC) values. The addition of CBC values moderately increased model performance (Supplementary Fig. 9), but the difference was not significant, and the test dataset was small ($n = 163$). When blood biomarkers were incorporated, decreased platelet count was an important predictor of *Borrelia*.

## Discussion

In this study, we employed untargeted RNA–mNGS and targeted 16S sequencing to understand the causes of acute febrile illness at an ambulatory clinic in Thiès, Senegal. As the first unbiased investigation of causes of fever in Senegal, our findings highlight the importance of looking across many pathogen types simultaneously to understand their contributions—both individually and as co-infections—to disease and enable comparison of the strengths of genomic tools as well as clinical and epidemiological data to support disease characterization.

Our results confirm that arthropod-borne bacterial pathogens, particularly relapsing fever *Borrelia*, are the major identifiable causes of NMFI but remain underdiagnosed. The frequency of these pathogens is broadly consistent with previous findings in targeted studies from other regions of Senegal[17–20]. *Borrelia spp.* associated with human and zoonotic infection have been identified in ticks across West Africa[16,36,40,41], but human surveillance has not been done.

Our broad approaches and sampling illuminated co-infections and missed diagnoses that may be clinically relevant. Both *Borrelia* and *Rickettsia* occurred frequently as co-infections in our study. *Borrelia/Plasmodium* co-infections have been previously detected in Senegal[42], and murine model evidence suggests co-infection increases the risk of severe malarial illness[43]. *Rickettsia* was detected more often as a co-infection with *Plasmodium* ($n = 3$) or *Borrelia* ($n = 2$) than as a stand-alone infection ($n = 3$). *Plasmodium/R. felis* co-infections have been previously identified in Senegal[44], and *Anopheles gambaie*, the primary malaria vector in Senegal, may be able to transmit *R. felis*[45]. Alternatively, Labruna and Walker proposed that *R. felis* may not be a pathogen itself but rather a symbiont of parasites that infect humans, such as protozoa and helminths[35]. Whether *Rickettsia* is contributing to pathogenesis or simply co-transmitting in the *Rickettsia/Plasmodium* co-infections detected in this study remains an open question. More research is needed to understand the interaction between *Plasmodium* and *R. felis*. Given the common occurrence of bacterial/*Plasmodium* co-infections and the potential for negative outcomes without proper treatment of both pathogens, it is important that both surveillance and diagnostic approaches do not stop at detection of the first pathogen.

Despite using multiple methods and searching across kingdoms, we did not find any pathogen in over 70% of NMFI cases in this study. Although our enrollment criteria focused on suspicion of malaria, many of these patients reported sore throat and difficulty breathing (2019: 35%, 40/113) or abdominal pain (4.4%, 5/113). Given that only cell-free plasma was sequenced, our approach may have missed infections restricted to other body compartments, including the respiratory tract and digestive tract, and may have limited our ability to detect intracellular pathogens. Though we did detect some intracellular pathogens (e.g., *Rickettsia*), we did not observe other intracellular bacterial pathogens, particularly *Salmonella spp.*, which are a common cause of febrile illness in hospitalized patients in Sub-Saharan Africa[12,13,46]. Studies from elsewhere in Sub-Saharan Africa suggest a lower incidence of bacteremia in outpatients consistent with our



a

| Model type:<br>Comparison group(s):<br>Training set:<br>Testing set: | Clinical only<br>Other NMFI<br>2018-2019 (n = 451)<br>2018-2019 (n = 451) |
|---|---|
| **Recall**<br>**(95% CI)** | 0.861<br>(0.837-0.910) |
| **Precision**<br>**(95% CI)** | 0.792<br>(0.696-0.892) |
| **F1**<br>**(95% CI)** | 0.823<br>(0.766-0.900) |
| **AUC/ROC**<br>**(95% CI)** | 0.911<br>(0.872-0.925) |
| **AUC/PR**<br>**(95% CI)** | 0.872<br>(0.806-0.910) |

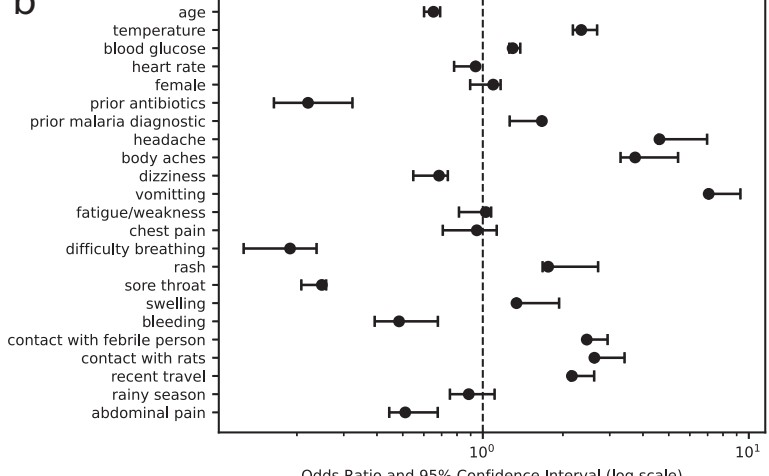

**Fig. 4 | Results of best performing weighted logistic regression model to distinguish *Borrelia* infection from other NMFI. a** Model performance metrics of best performing weighted logistic regression model to distinguish *Borrelia* infection from other NMFI using clinical data, including demographics, symptoms, exposures, and vital signs (trained on 2018-2019 data, tested on the same data with bootstrapping and cross-validation, $n = 451$) and **b** mean odds ratios (center point) and 95% confidence interval (error bars) for all features retained in the final model after feature selection. Source data are provided as a Source Data file.

study[12,47–49]. however, we cannot rule out that other intracellular bacterial infections were present at low titers in the blood and, therefore not detected by plasma 16S and RNA–mNGS. Additionally, fever can persist beyond the acute viremic or bacteremic phase, limiting our ability to detect the causative pathogen. Lastly, noncommunicable diseases, such as cancer or autoimmune disease, can also cause fever. In clinical samples, negative mNGS results are common, but it can be difficult to determine whether these represent false negatives due to limited sensitivity or true negatives[7]. As the burden of noncommunicable disease rises in West Africa[50], the approach to diagnosing febrile illness must consider non-infectious etiologies.

Our results also demonstrate the difficulty of using RNA–mNGS to detect pathogens with different nucleic acids and cellular compositions, limiting our ability to do truly comprehensive surveillance. Several of the bacterial infections in this study population, particularly those with low abundance in the blood, were detected by 16S but missed by RNA–mNGS. Similarly, we were not able to resolve putative fungal hits to a suitable level of taxonomic resolution; to capture fungal pathogens, extraction methods should be optimized for lysis of fungal cells, and more targeted methods such as ITS sequencing could be used. We observed wide variations in the number of *Plasmodium* reads among RDT-positive patients and a weak correlation with parasite density, suggesting that factors beyond parasite burden affect the number of reads recovered. There are also still many challenges in interpreting RNA–mNGS data. In particular, RNA–mNGS is highly susceptible to contamination, such as the novel *Reoviridae* RdRp sequences we detected across febrile cases, healthy controls, and non-template controls. This type of contamination, which likely differs between sample types and labs, can only be evaluated when negative controls are processed alongside samples[51].

We find that combinations of clinical signs and symptoms can increase suspicion for *Borrelia* and support targeting diagnostic testing and clinical care. Due to the lack of available point-of-care tests, many clinics in LMICs rely on malaria RDTs as the primary diagnostic for febrile illness and may give blanket antibiotic treatment to RDT-negative patients[52]. Concerns have been raised about this practice[3], including driving antibiotic resistance and exposing patients without evidence of bacterial infection to unnecessary risks or side effects. Even in patients with a bacterial infection, tetracycline antibiotic treatment—the recommended therapeutic for *Borrelia* and *Rickettsial* infections—poses the risk of Jarisch–Herxheimer reaction (JHR), a severe inflammatory response to spirochete lysis[53]. Without accurate diagnosis of spirochetal infections, including *Borrelia*, patients and providers cannot effectively anticipate and monitor for this complication. We identify a number of clinical and immunological features that could bolster support for a *Borrelia* diagnosis. Our weighted logistic regression model indicates that when larger study populations and diverse data types are aggregated, *Borrelia* can be differentiated from other NMFI with reasonable accuracy. The addition of laboratory measures, such as CBC, did not substantially improve differential diagnosis. Further, data from a subset of our study population supports the hypothesis that a chemokine panel including TRAIL, IP-10, and CRP, previously employed for the detection of bacterial infections in hospitalized children[54], could apply more broadly and help distinguish bacterial infections such as *Borrelia* from other causes of febrile illness in the ambulatory setting. However, none of these approaches are currently actionable as point-of-care tests, and their lower precision would still necessitate confirmatory diagnostic testing.

The challenge of quickly and accurately differentiating *Borrelia* infections underscores the need for improved diagnostics. The diagnostic gold standard for *Borrelia* remains the microscopic examination of blood smears, which is highly dependent on a trained practitioner and has low sensitivity in our study and others[36]. Further, smears cannot distinguish species. Louse-borne relapsing fever agent *B. recurrentis* has a higher fatality rate and risk of JHR than tick-borne

relapsing fever species, but with smear alone, clinicians cannot distinguish which disease they are treating unless the history of vector exposure is known. In Lyme Disease, molecular surveillance has shown that different species have different clinical presentations and severities[55]. Species-specific surveillance at scale could reveal analogous differences between relapsing fever *Borrelia*. Our study showed that molecular methods, including 16S sequencing and mNGS, can detect and distinguish *Borrelia* infections but are not practical in the clinical setting. Increased availability of qPCR assays and improved point-of-care nucleic acid diagnostics could enable the detection and treatment of *Borrelia* across differently resourced clinical settings and improve our understanding of the geographic range and diversity of this pathogen.

Decreased malaria transmission coupled with changes in the range and incidence of other pathogens—influenced in part by global climatic changes—is shifting the burden of pathogens causing febrile disease[56]. New technologies have greatly increased our ability to detect both established and emerging pathogens. Fully understanding this landscape will require surveillance systems utilizing untargeted approaches that have been validated for viral, bacterial, and eukaryotic pathogens and diverse sample types. In tandem, for *Borrelia* and other pathogens known to frequently cause NMFI, the development and availability of diagnostic tests that are cheap, rapid and sensitive will be key to enable appropriate clinical treatment at the point of care and support deeper investigations of the pathophysiology of disease.

## Methods
### Sample collection
We performed a cross-sectional study of febrile and healthy individuals. Febrile cases were selected from patients presenting to the SLAP outpatient clinic in Thiès, Senegal, during the collection period under local IRB (SEN15/46) and Harvard IRB (IRB19-0023). During times when the study personnel were onsite, all patients who met the following inclusion criteria and gave consent were enrolled: (1) Febrile symptoms within the 3 days up to and including the day of presentation, (2) Age 2–75 years, and (3) Ambulatory with no signs of severe malaria (glucose <2.2 mM, hemoglobin <5 gms/dL). Healthy controls were recruited via a call for participants. All volunteers who met the following inclusion criteria and consented were enrolled: (1) No febrile symptoms within the 3 days up to and including the day of presentation and (2) Age 2–75 years. In each season, enrollment continued until approximately the desired number of participants was reached. In order to detect pathogens present at 1% or greater prevalence, we aimed for 200 febrile cases and 200 healthy controls in each season in 2018. In 2019, to avoid the overrepresentation of healthy individuals, we aimed for 100 febrile cases and 50 healthy controls in each season.

Informed consent was obtained for all enrollees (febrile and healthy); for minors under 18 years of age or individuals unable to provide their own consent, the consent of a parent or legal guardian was obtained. The study team explained to potential participants that their participation was strictly voluntary and that they could withdraw from the study at any time without any penalties or consequences, and translations were made for potential participants (or their parent/legal guardian) who do not understand or cannot read the language in which the consent form was produced. A subset of DENV-positive samples from 2018 were previously sequenced and published[14] and were excluded from this study other than being used as a viral outgroup in the chemokine/cytokine analyses. At the time of enrollment, a structured interview including personal information, demographics, and self-reported symptoms was completed, vital signs were measured, and blood was drawn by a trained practitioner. *P. falciparum* RDT (Bioline Malaria Ag P.f., Abbott), thick blood smear, and thin blood smear were performed, and blood glucose (HemoCue 201/301) and hemoglobin (HemoCue Glucose 201) were measured.

## Sample processing

Blood samples were stored on ice upon collection (maximum 8 h). In the lab, blood samples were split into two aliquots. In total, 200 μl of the first aliquot was used for an automated blood count with differential (Mindray BC 20 s), and 200 μL was used for a blood spot (Whatman filter paper). The remaining aliquot was centrifuged at 800× g for 10 min to separate plasma, buffy coat, and red blood cells. 140 μL of plasma was transferred to a tube containing 560 μL of inactivation buffer (AVL, Qiagen). The remaining plasma was stored without inactivation. Red blood cells, buffy coat, plasma, and inactivated plasma were flash-frozen and stored at −80 °C until further processing. Total nucleic acids were extracted with Qiagen QIAmp Viral Mini RNA according to the manufacturer's protocol. Total nucleic acids were split into two aliquots and treated with Lucigen RNAse I or Ambion TURBO DNase to obtain purified DNA and RNA, respectively. DNA was extracted from dried blood spots with QIAmp DNA Blood Mini Kit (QIAGEN) according to the manufacturer's protocol.

## Untargeted RNA sequencing

**Library preparation and sequencing.** Untargeted RNA sequencing libraries were prepared as described previously[57]. Briefly, cDNA synthesis was performed from DNAse-treated RNA using random primers (Invitrogen) and SSIV. Libraries were prepared from cDNA using Nextera XT (Illumina) and UD index primers (IDT/Illumina) with the following modifications to account for low cDNA input: the volume of ATM was decreased from 1 μL to 0.5 μL per 10 μL reaction and PCR cycles were increased to 17 cycles. Libraries were purified using AMPure XP (Agilent), quantified using KAPA Biosystems Universal Library Quantification, and pooled equally for 75 bp paired-end sequencing on NovaSeq SP to obtain at least 2 million reads per sample.

**Taxonomic classification of RNA–mNGS reads.** Untargeted RNA sequencing reads were processed using viral-ngs v2.1.33.16 (https://github.com/broadinstitute/viral-ngs)[58]. Briefly, reads were demultiplexed, adapter sequences were trimmed, sliding-window quality filtering was performed, and human reads were filtered out using the viral-ngs demux_only [https://dockstore.org/workflows/github.com/broadinstitute/viral-pipelines/demux_only:master] and classify_single [https://dockstore.org/workflows/github.com/broadinstitute/viral-pipelines/classify_single:master?tab=info] workflows. Cleaned reads were uploaded to NCBI SRA under BioProject PRJNA662334 (Accession numbers: SRR24622550-SRR24622641, SRR24995052-SRR24995258). Samples with insufficient reads (<2 million) were removed from further analysis (n = 1, SHC1064, Accession: SRR24995248).

Taxonomic classification of human-depleted reads was performed using Kraken2[59] with the PlusPF database (https://benlangmead.github.io/aws-indexes/k2, downloaded 12-13-2022). Kraken2 results were thresholded to consider only viral genera with at least 5 reads/million raw reads. Results were filtered to remove viral taxa detected in non-template controls (at least one non-template control) and healthy patients (at least 2 healthy patients). Further, viral genera with non-vertebrate hosts were filtered out.

To verify Kraken2 classifications, we performed a protein-sequence similarity search using DIAMOND v2.0.15[60]. Cleaned, de-duplicated reads were used for de novo contig assembly with SPAdes and DIAMOND-blastx was run on all de novo contigs >100 bp long with the complete nr database (downloaded December 2022). The least common ancestor of the top e-value hits was identified using a custom script (lakras/bio-helper-scripts/blast/retrieve_top_blast_hits_LCA_for_each_sequence.pl). A classification was considered verified by DIAMOND if the cumulative length of contigs with a high identity match (mean identity of top e-value hits ≥ 95%) to the expected viral genus was at least 1 kb. After DIAMOND-blastx verification, raw reads were aligned to the NCBI Virus RefSeq and assessed for evenness and depth of coverage.

To assess for the presence of divergent viral taxa that are not well represented in the database and may be missed by Kraken2, DIAMOND-blastx results were filtered for viral hits with low identity (20–80% amino acid identity). Results were filtered for viral families with a per-sample cumulative de novo contig length ≥ 500 bp. *Narnaviridae* and *Reoviridae* sequences were aligned with all available RdRp sequences for each family in nr using MAFFT v1.5.0 and trimmed with trimAl[61]. Maximum likelihood phylogenetic trees were generated in IQ-TREE v1.6.12[62] with bootstrapping (n = 1000) and visualized in FigTree v1.4.4[63].

**Viral genome assembly and phylogenetic analysis.** For virus-positive samples, reference-guided de novo assembly was performed using viral-ngs v2.1.33.16[58] assemble de novo workflow [https://dockstore.org/workflows/github.com/broadinstitute/viral-pipelines/assemble_denovo:master?tab=info]. The NCBI Virus RefSeq, as well as any >80% complete genomes available from Senegal, were provided as references. For samples with near-complete or complete (>80%) genomes, genotype was determined with GenomeDetective (https://www.genomedetective.com/, DENV, HBV) or by the phylogenetic tree (Parvovirus B-19, HPgV-1). Phylogenetic trees were generated with all >80% complete references from Africa downloaded from NCBI viruses (DENV1, DENV3, HBV), or all global >80% complete genomes available in NCBI virus (Parvovirus B-19, HPgV-1). Sample sequences and reference sequences were aligned using MAFFT v1.5.0 v7, and alignments were trimmed with trimAl v1.4.rev15 to remove any bases with coverage in <80% of sequences. Maximum likelihood phylogenetic trees were generated in IQ-TREE v1.6.12 with a GTR-gamma substitution model and bootstrapping (n = 1000). Visualizations were generated with FigTree v1.4.4[63].

**Plasmodium RNA–mNGS analysis.** In order to quantify *Plasmodium* rpm, the number of reads classified as *Plasmodium* at the genus level by Kraken2 was divided by the raw read count in millions. The cut-off for considering a sample positive for *Plasmodium* by mNGS was determined by calculating the 99th percentile for healthy control samples, 550 *Plasmodium* rpm. Cleaned deduplicated reads were aligned to the *P. falciparum pfhrp2* [https://plasmodb.org/plasmo/app/record/gene/PF3D7_0831800] with the viral-ngs align_and_count workflow [https://dockstore.org/workflows/github.com/broadinstitute/viral-pipelines/align_and_count_multiple_report:master].

**Fungal RNA–mNGS analysis.** In order to detect possible fungal infections, we calculated the rpm classified as Fungi by Kraken2 at the kingdom level and determined the 99th percentile for healthy controls, 3399 rpm. We identified 5 samples (4 febrile, 1 control) with high fungal reads. We attempted taxonomic classification of these reads with Kraken2 at the genus level and saw hits across multiple genera for many samples (Supplementary Fig. 2b). To try to improve fungal classification, we performed a nucleic acid search of all de novo contigs from these 5 samples with megablast against nt (Supplementary Fig. 2b) and a translated nucleic acid search with DIAMOND-blastx (as described above). For both searches, we found the least common ancestor of the top hits (as described above) and filtered hits in the kingdom Fungi (taxid: 4751) with >90% identity and >30% coverage.

## PCR and qPCR assays

All qPCRs were performed on the QuantStudio 6 Flex ReadTime PCR (Applied Biosystems). Primer sequences and sources are described in Supplementary Table 4. All qPCRs were performed with a standard curve using a synthetic DNA standard (see Supplementary Table 5). Samples were tested in triplicate and considered positive if all three wells had a cycle threshold of less than 40.

Total bacterial load was quantified from extracted plasma DNA with primers targeting the V1–2 region with Power SYBR Green PCR

MasterMix (Thermo Fisher) under the following conditions: 300 nM each V1–2-F/R primer, 95 °C hold for 10 min, 40 cycles of 95 °C for 15 s, 50 °C for 1 min, 75 °C for 30 s. Pan-*Borrelia* qPCR was performed on DNA extracted from plasma with Power SYBR Green PCR MasterMix (Thermo Fisher) under the following conditions: 400 nM each *Borrelia*-F/R primer, 95 °C hold for 10 min, 40 cycles of 95 °C for 15 s, 60 °C for 1 min. Pan-*Plasmodium* qPCR was performed on DNA extracted from dried blood spots (DBS) with Power SYBR Green PCR MasterMix (ThermoFisher) under the following conditions: 600 nM each Spp-F/R primer, 95 °C hold for 10 min, 40 cycles of 95 °C for 15 s, 60 °C for 1 min. *Plasmodium falciparum* specific qPCR was performed on DNA extracted from DBS with TaqMan Fast Advanced Master Mix (Applied Biosystems) under the following conditions: 450 nM each Fal-F/R primer and 125 nM Fal probe, 95 °C hold for 20 s, 40 cycles of 95 °C for 1 s, 60 °C for 30 s on the fast setting.

Amplification of pfhrp2 exon 2 was performed on DNA extracted from DBS with Q5 HighFidelity 2× Master Mix (NEB) under the following conditions: 500 nM each Pf3D7_0831800-F/R primer, 98 °C hold for 30 sec, 40 cycles of 98 °C for 10 s, 60 °C for 30 s, 72 °C for 30 s, followed by 72 °C hold for 2 min. Amplification of the flanking gene was performed with Q5 HighFidelity 2× Master Mix (NEB) under the following conditions: 500 nM each Pf3D7_0831700-F/R, 98 °C hold for 30 s, 40 cycles of 98 °C for 10 sec, 64 °C for 30 s, 72 °C for 30 s, followed by 72 °C hold for 2 min. PCR products were visualized on E-Gel EX 2% Gel (Thermo Fisher) with the E-Gel 1 kb plus DNA ladder (Thermo Fisher), and images were captured with FluorChem E imager (ProteinSimple).

## 16S sequencing

**Library preparation and sequencing.** Samples with high bacterial load (V1–2 qPCR CT < 31.5) were selected for 16S sequencing (2019: febrile $n = 129$, healthy $n = 35$, 2018: febrile $n = 84$). Libraries were generated by amplification with tailed universal primers Tail-V1-2F and Tail-V1-2R (Supplementary Table 4) targeting variable regions 1–2. Amplification was performed on 5 μl of RNAse-treated DNA using Q5 high fidelity polymerase (NEB) with forward and reverse primers at 100 nM each with the following cycling conditions: 98 °C for 30 s; 35 cycles of 95 °C for 15 s and 63 °C for 2 min, 4 °C hold. PCR products were cleaned using Ampure XP (Agilent) with a 0.7× ratio to remove primer dimers. Adapters and barcodes (BroadDuplex Seq) were added with a second PCR reaction using Q5 polymerase and the following cycling conditions: 98 °C for 30 s; 18 cycles of 95 °C for 15 s, 60 °C for 15 S, 72 °C for 30 S, followed by 72 °C for 5 min, 4 °C hold. A subset of libraries was visualized with hsDNA BioAnalyzer (Agilent), and all libraries were quantified with KAPA Biosystems Universal Library Quantification Kit before pooling equally across samples and 250 bp paired-end sequencing on Illumina MiSeq v2 with 40% PhiX, given the low diversity of the single-amplicon library.

**Analysis.** Sequencing reads were demultiplexed using viral-ngs v2.1.33.16[58], and demultiplexed FASTQ files were imported into qiime2 v2022.2.0[64] for further analysis. Demultiplexed reads were uploaded to NCBI SRA under BioProject PRJNA662334. Briefly, after sliding window quality filtering and adapter trimming with cut-adapt (minimum-length 20), paired reads were joined with vsearch (tuncqual 15, minlen 35, minovlen 10, maxdiffs 3), and ASVs were generated with Deblur (trim-length 280). ASVs were taxonomically classified by blastn search of ASV sequences against the NCBI 16S rRNA db (downloaded June 2023). The least common ancestor of the top e-value hits was determined using a custom script (lakras/bio-helper-scripts/blast/retrieve_top_blast_hits_LCA_for_each_sequence.pl). Results were filtered to include only ASVs that had identity ≥94.5% to the closest hit, as this has been proposed as a rational cutoff for 16S rRNA diversity within genera[65]. Abundance was calculated by dividing the number of ASVs in each genus by the total number of ASVs in each sample.

Abundance data was filtered to show only taxa accounting for greater than 5% of ASVs. Taxa detected in non-template controls ($n \geq 1$) and healthy controls ($n \geq 2$) were filtered out.

For *Borrelia* and *Rickettsia* positive samples, sample ASVs were aligned with MAFFT v1.5.0[66] (Geneious v2022.0.2) with curated contextual sequences for relapsing fever *Borrelia* spp. and *Rickettsia* spp. known to infect humans. When available for a given species, the high-quality 16S sequencing from the NCBI 16S rRNA BioProject [https://www.ncbi.nlm.nih.gov/bioproject/33175] was used. For species not included in the NCBI 16S rRNA BioProject, all available sequences from the SilvaNRRef small subunit database were downloaded, sequenced with complete coverage of the v1–2 region were selected, and a representative sequence was selected randomly for each species. Alignments were trimmed manually in Geneious v2022.0.2 to include only the v1–2 region, and a maximum-likelihood phylogenetic tree was generated using IQ-TREE v1.6.12 with bootstrapping ($n = 1000$) and annotated with FigTree v1.4.4[63].

## IGS sequencing

The intergenic spacer (IGS) region was amplified using a nested-PCR. First, the region was amplified from 5uL of extracted DNA with primers targeting the IGS region (IGS-outer F/R, see Supplementary Table 4) using Q5 polymerase and the following cycling conditions: 98 °C for 30 s; 35 cycles of 94 °C for 30 s, 66 °C for 30 S, 74 °C for 60 s, followed by 74 °C for 2 min, 10 °C hold. After a 0.75× AMPure XP cleanup to remove excess primer, a nested PCR was performed (IGS-inner F/R, see Supplementary Table 4) using Q5 polymerase and the following cycling conditions: 98 °C for 30 s; 30 cycles of 94 °C for 30 s, 67 °C for 30 S, 74 °C for 60 s, followed by 74 °C for 2 min, 10 °C hold. Amplified product was cleaned with AMPure XP (0.75×), visualized with BioAnalyzer TapeStation and bi-directional Sanger sequencing was performed (Azenta). Paired Sanger sequencing traces were analyzed in Geneious v2022.0.2 to generate a consensus sequenced and aligned with all available reference sequences for *B. crocidurae, B. duttonii, and B. recurrentis* using MAFFT v1.5.0. A maximum-likelihood phylogenetic tree was generated using IQ-TREE v1.6.12 and annotated with FigTree v1.4.4[63].

## Blood smear examination

Thick and thin blood smears were fixed with methanol and stained with 3 mL 10% Giemsa for 10 min. Stained smears were examined under a brightfield microscope for evidence of *Borrelia*. Slides were first scanned at low (40×) magnification and then at high (100×) magnification with oil immersion to quantify organisms per field (representative image, Supplementary Fig. 7b).

## Serology

We assessed levels of common cytokines and chemokines using the Luminex platform on plasma from a subset of patients and controls. Specifically, we selected 10 healthy controls and 30 cases representing 10 each with a confirmed diagnosis of; malaria (based on malaria RDT administered at enrollment), *Borrelia* (based on pan-*Borrelia* qPCR[38]), or viral infection. Samples with a viral infection were all confirmed to have Dengue virus and are a subset of those cases previously described elsewhere[14]. Cases were selected randomly within those meeting each criteria. All samples were collected in 2018. We used the Inflammation 20-Plex Human ProcartaPlex™ Panel (Invitrogen) according to the manufacturer's instructions. Two of the assays (IFN-gamma and GM-CSF) did not generate results due to errors during data collection. We additionally performed a 2-plex assay for Human CRP and TRAIL (using CRP and TRAIL ProcartaPlex™ simplex kits from Invitrogen); two markers previously found to differentiate bacterial infections in hospitalized children[54], as these were not included on the predefined panel. All samples were run in duplicate on a Luminex MAGPIX® instrument in Senegal. We confirmed that standards showed the

expected values and calculated the average across duplicates, which we reported in the text.

## Multivariate model

We developed a weighted logistic regression model that considered 46 candidate predictors based on clinical symptoms, vital signs, and key demographic information to predict *Borrelia* infection. Before modeling, the dataset underwent the following preprocessing steps: imputation of missing values with the Multivariate Imputation by Chained Equations (MICE) algorithm from the fancyimpute[67] Python library, one hot-encoding of categorical features, and standardization of continuous features using the StandardScaler from the scikit-learn library. Given that normal heart rate varies significantly with age, heart rates were binned and treated as a categorical variable. Heart rate categories were high, low, or normal based on the Pediatric Advanced Life Support guidelines as follows: for ages 3–5, normal range of 80–120; ages 6–11, normal range of 75–118; for age 12+, normal range 60–100[68].

The sample size was relatively small, with 526 patients and only 38 q-PCR-confirmed cases of *Borrelia* infection. Given this inherent class imbalance in the dataset, a combination of techniques was employed to address this challenge. The Synthetic Minority Oversampling Technique (SMOTE) was applied to generate synthetic samples for the minority class (q-PCR confirmed *Borrelia* case)[69]. This technique mitigates class imbalance by interpolating between existing samples and creating synthetic instances of the minority class. During model training, class weights were also applied to adjust the influence of each class. The class weights were inversely proportional to the class frequencies, with higher weights assigned to the minority class.

A logistic regression model was chosen for its interpretability and ability to estimate the probabilities of class membership. After a univariate screening for statistically significant predictors using a chi-squared test ($p$-value < 0.05), several Feature Selection techniques were applied to construct the final set of predictors in the model. Before training the model, feature importance was assessed using a combination of three different feature selection methods; mutual information classification[70], Recursive Feature Elimination[71], and Lasso L1 regularization[72,73] to ensure consistent elimination of redundant variables. The selected features were used in the final weighted logistic regression model to analyze the performance of *Borrelia* prediction. The regularization parameter from Lasso was tuned using cross-validation of different alpha values, and the best alpha value was used to train the final model and evaluate its performance on the test set.

Model performance was evaluated with bootstrapping and five-fold cross-validation to quantify optimism and generalizability. The optimism-corrected metrics include F1-score, precision, recall, area under the ROC curve (AUC-ROC), and area under the precision-recall curve (AUC-PR). Odds ratios and their corresponding confidence intervals for all the model coefficients were calculated to assess the impact of predictor variables on the odds of the outcome. All predictive analyses were conducted using the Python programming language. The scikit-learn library was utilized for preprocessing, model training, and evaluation. Statistical analyses and visualization of odds ratios and confidence intervals were done using the R statistical package.

## Inclusion and ethics statement

This study was designed, implemented, and analyzed in close collaboration between US-based researchers and Senegalese researchers at Cheikh Anta Diop University/CIGASS, the SLAP outpatient clinic, and the National Malaria Control Program. All study team members are included as authors, and their contributions are detailed in the CRediT statement. The protocol was approved by the Harvard IRB as well as the IRB of the Ministry of Health in Senegal, and all co-authors who worked directly with samples in Senegal or the U.S. received appropriate biosafety training. Sequencing and analysis were conducted at the Broad Institute due to the higher capacity for high-depth sequencing. Sample aliquots are stored at both CIGASS and the Broad Institute, and all sequencing data and sample-associated data are accessible to study teams at both locations. Relevant research from other local teams is cited as appropriate.

## Reporting summary

Further information on research design is available in the Nature Portfolio Reporting Summary linked to this article.

## Data availability

RNA−mNGS short read sequence data and 16S v1−2 amplicon sequence data have been deposited in NCBI's SRA databases under BioProject PRJNA662334. Viral genomes have been deposited in NCBI's Genbank (see Supplementary Table 2 for BioSample identifiers). Source data are provided in this paper.

## Code availability

The Python source code for the multivariate Borrelia prediction model is available at github.com/colabobio/borrelia-diagnosis-prediction-models. The custom scripts used to retrieve the least common ancestor of top blast/DIAMONd hits are available at lakras/bio-helper-415 scripts/blast/retrieve_top_blast_hits_LCA_for_each_sequence.pl.

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

## Acknowledgements

We thank the Program National de Lutte contre le Paludisme (National Malaria Control Program) for providing malaria surveillance data and Fama and Astou for support with data entry and organization. We thank Lydia Krasilnikova for advice and custom scripts for BLAST analysis and Flavia Negrete for support with data uploads and sharing. We thank Jacob Lemieux and Gordon Adams for providing valuable guidance on *Borrelia* sequencing. Support for this project was provided in part by the National Institutes of Allergy and Infectious Diseases through the H3Africa and Center for Research in Emerging Infectious Disease programs (U54HG007480, U01HG007480, and U01AI151812 to C.T.H. and P.C.S.), and the Genomic Center for Infectious Disease (U19AI110818 to P.C.S.). This work is made possible by support from Flu Lab and a cohort of generous donors through TED's Audacious Project, including the ELMA Foundation, MacKenzie Scott, the Skoll Foundation, and Open Philanthropy. Z.C.L was supported by the National Institute of General Medical Sciences (T32GM007753 and T32GM144273). K.J.S. received support from the American Society of Tropical Medicine and Hygiene Centennial Award. P.C.S. is an investigator supported by the Howard Hughes Medical Institute (HHMI). The content is solely the responsibility of the authors and does not necessarily represent the official views of the National Institute of General Medical Sciences, National Institutes of Allergy and Infectious Diseases, or the National Institutes of Health.

## Author contributions

According to the CRediT format, author contributions are as follows: Conceptualization: A.B.D., A.S.B., Z.C.L., N.S., P.C.S., K.J.S., and D.N.; Methodology: A.B.D., A.S.B., Z.C.L., N.S., P.C.S., K.J.S., and D.N.; Software: D.J.P.; Validation: P.C.S., K.J.S., and D.N.; Formal analysis: Z.C.L., K.J.S., W.M., and A.C.; Investigation: Z.C.L., A.S., A.G., T.N., M.S., Y.D., A.M., and I.M.N.; Data Curation: J.G., M.Ndiop, D.S.; Writing—original draft: Z.C.L. Writing—review & editing: K.J.S., S.F.S., B.L.M, and P.C.S. with input from all authors; Visualization: Z.C.L., W.M.; Supervision: P.C.S., K.J.S., and D.N.; Project administration: M.F.P., M.Ndiaye; Funding acquisition: P.C.S., D.N., K.J.S., and D.J.P.

## Competing interests

P.C.S. is a co-founder of, shareholder in, and consultant to Sherlock Biosciences, Inc. and Delve Bio, as well as a Board member of and shareholder in Danaher Corporation. The remaining authors declare no competing interests.
