## [Peer Review File · Nature Communications]

Investigating the etiologies of non-malarial febrile illness in Senegal using metagenomic sequencingReviewers' comments:

Reviewer #1 (Remarks to the Author):

The stated objective of this study was to determine and characterize the major causes of NMFI in Thiès, Senegal. This objective has not been achieved. The causes of severe febrile illness were not studied, since only outpatients were included. The diagnostic tests used involved the detection of pathogens in plasma samples by sequencing. This method lacks sensitivity for the diagnosis of intracellular bacterial infections such as *Salmonella* species, none of which were found in this study, whereas previous studies have shown they are a major cause of febrile illness in West Africa 1. The most commonly identified infections were *Borrelia*, which was found in 18/204 febrile patients (9%) and Rickettsial infections, both of which have been previously shown to be common causes of fever in Senegal. Microscopy of blood film, the current gold standard for the diagnosis of *Borrelia* infection, was shown to lack sensitivity compared to molecular methods but, since these are not widely available, the authors developed a weighted logistic regression model that considered 46 candidate predictors based on clinical symptoms, vital signs, and key demographic information to predict *Borrelia* infection, but it seems unlikely that this will be widely used at the primary care level, where a trial of treatment with tetracycline would be easier to deliver in a case of suspected borreliosis.

1. Reddy EA, et al. *Lancet Infect Dis* 2010; 10: 417

Reviewer #2 (Remarks to the Author):

The manuscript "Improving diagnosis of non-malarial fevers in Senegal: *Borrelia* and the contribution of tick-borne bacteria" is a well written and informative description of identification and sequence analysis of non-malarial pathogenic organisms identified in human specimens. The authors describe findings from two sequence based approaches that highlight common causes of febrile illness that may not be tested for or standard diagnostic approaches may miss. The authors highlight findings of relapsing fever group borrelia as a significant cause of febrile illness in the subset of specimens tested. This study highlights how sequence based approaches to clinical identification of infections can inform patient care and differential diagnosis of patients seeking care for febrile illness with a complex genetic diversity of potential infecting pathogens. The manuscript reads well, and I only had a few minor points of clarification I would like to see the authors address.

Lines 80-82: It is unclear here if co-infections were identified between bacterial and viral positive specimens. Were the virus positive specimens tested for co-infection by 16S? Can you make more clear if this was the case. Were no viral/bacterial infections identified? How was this tested specifically?

Supplementary figure 7A is supposed to show the placement of the 16S V1V2 sequences from *Borrelia* positive specimens amongst other relevant borrelia species. The tree includes numerous species from the outlying "Lyme Disease" grouping. The majority of the species listed have never been shown to cause Lyme disease and, as labeled, the figure is very misleading. The figure should include minimal "Lyme group" species as it is an outgroup, and many more of the ~20 relapsing fever borrelia species. This would help the reader understand how 16S could potentially differentiate from other clinically relevant relapsing fever borrelia as V1V2 16S sequence should be sufficient enough to differentiate *B. crocidurae* from most others. This placement should then be compared to the confirmatory IGS, 23S sequencing presented in Fig 2C that include *B. duttonii* and *B. recurrentis*.

Lines 179-183: The term "isolate" is used throughout this paragraph. Suggest replacing this term with a more accurate description used for culture free approach employed here. Unless cultured organisms were recovered for these six specimens, "isolate" seems inaccurate.

Lines 205-212: Can the authors comment on whether qPCR was more sensitive than 16S sequencing based on these results in their hands? Did the additional 5 borrelia positive specimens that did not undergo 16S sequencing also not have RNA-mNGS data showing bacterial infection? I was unable to find supplementary figure 7E mentioned in the text for these specimens.

Lines 328-329: If symptomology and blood marker based differential criteria are not clinically

actionable could this information still be disseminated for resource limited point-of-care clinicians to avoid potential JHR in RDT negative patients given RF borrelia were so frequently identified? This last sentence suggests the real issue is needing better point of care diagnostics for RF-borrelia. Does the symptomology differential lead to better test ordering?
Lines 330-338: Similar to the point above...The limitation of blood smears to differentiate borrelia species seems irrelevant to the point of this paragraph because RT-PCR does not necessarily provide species level resolution. The blood smear appeared to perform poorly compared to direct detection methods in this study and that seems more relevant. Genus specific RT-PCR seemed to perform quite well once an appropriate assay was used based on what the sequencing based approaches found. Please clarify the necessity of species level diagnostic resolution, if any, as it pertains to patient treatment and the feasibility of frontline diagnostic approaches and which ones would improve the current situation.

Reviewer #3 (Remarks to the Author):

This interesting manuscript by Levine and colleagues reports on a RNA-metagenomic sequencing survey of blood samples of Senegalese febrile patients that were considered to not have malaria. The continued burden of fever in sites where antimalarial interventions are successful provide an excellent opportunity to discover other infections that might have been masked by the default malaria diagnosis; and, the use of modern diagnostic methods will potentially expand our knowledge of the association of other infectious agents with human disease. Although there is much emphasis on the confirmation of borreliosis as a cause of nonmalarial fevers, the finding is not new and indeed the literature is only selectively cited. Ndiaye et al 2021 reported 12% of 800 febrile samples contained evidence of a relapsing fever borrelia in another part of Senegal. Similarly, although rickettsiosis is mentioned in multiple places in the manuscript as potentially burdensome (e.g., lines 164, 275, 281 et seq), there is again selective citation; Socolovskii et al. 2010 would be relevant for another area of Senegal, in which 8/134 nonmalarial febrile samples were considered to have evidence of infection by *R. felis*. (Of course, there is controversy regarding whether *R. felis* is actually a human pathogen, see Labruna and Walker 2014; this controversy should be discussed.) The phylogeny in supplemental figure 7B requires the inclusion of *R. felis*; it is not clear why the authors chose not to include such a common and well known *Rickettsia* sp in an alignment and analysis of rickettsial sequences. It is noteworthy that these febrile samples were so "clean". Indeed, only 1 HIV positive and 2 hepatitis B positives were found, perhaps testament to good educational outreach by local healthcare workers? It seems to me that this is the real story that needs to be emphasized, that state of the art methods (the caveats about sensitivity of mNGS for intracellular agents within blood samples notwithstanding; do note that SFG rickettsiae may be detected within circulating endothelial cells sloughing off of vasculitic lesions into the blood) reveal a lack of association between known or even putative infectious agents and nonmalarial fevers. The very conservative discussion about possible detection of commensals is very appropriate and highlights the challenge of this kind of work. Perhaps mNGS is not necessarily going to provide us with etiology for all fevers caused by infectious agents, and clearly will not be useful for noninfectious causes (malignancy, heatstroke, inflammatory conditions, among others). Some speculative discussion might be useful to inform the general reader of other potential causes of fever. Why do we usually just focus on infection as a cause of fever? This has implications for primary care and for understanding the global burden of disease.

Response to Reviewers, Nature Communications Appeal

We would like to thank the reviewers for their constructive feedback on the work and respond to each specific point in turn below.

Reviewer #1 (Remarks to the Author):

The stated objective of this study was to determine and characterize the major causes of NMFI in Thiès, Senegal. This objective has not been achieved. The causes of severe febrile illness were not studied, since only outpatients were included.

We acknowledge that NMFI incorporates both outpatient and hospitalized populations and agree that hospitalized patients are an important patient population. We, however designed our study specifically to enroll patients at the primary care level for several reasons:

- While there are high quality studies of hospitalized patients to help us understand severe disease, the primary care clinic is a critical site for pathogen surveillance for which there is little work being done.
- The vast majority of episodes of febrile illness will not rise to the severity of hospitalization, yet these cases are more likely to go undiagnosed given the limited diagnostic capacity at primary care clinics, particularly in West Africa.
- In our study, we aimed to determine the causes and clinical presentations of ambulatory febrile illness in order to better understand the range of pathogens circulating at the community level.
- Ours is one of the first large-scale and unbiased studies to carry out this investigation at a primary care center and the first in West Africa. Our results reveal important information on circulating pathogens and their clinical manifestations as well as highlighting the limitations of our current methods, even the most unbiased approach, for explaining all febrile illness.

We thank the reviewer for bringing attention to our lack of clarity in our motivation and study design, and we have modified the introduction to emphasize our focus on the primary care setting and the motivation for studying ambulatory patients. Specifically, we have included the following text in Introduction

- Line 28-30: *“In the absence of comprehensive surveillance efforts that capture multiple pathogen types, appropriate public health interventions are hindered by our limited understanding of the full landscape of common causes of NMFI at the community level.”*
- Line 49-52: *“Community and clinic based studies of patients with fever have revealed bacterial zoonoses are a common cause of ambulatory febrile illness in Senegal, including tick borne relapsing fever, Rickettsioses, Q fever, and Bartonella (Vial et al., 2006; Socolovschi et al., 2010; Sokhna et al., 2013; Mediannikov et al., 2014; Abat et al., 2016; Ndiaye et al., 2021). However, no unbiased surveillance of outpatient febrile illness has been done in Senegal to date.”*

The diagnostic tests used involved the detection of pathogens in plasma samples by sequencing. This method lacks sensitivity for the diagnosis of intracellular bacterial infections such as *Salmonella* species, none of which were found in this study, whereas previous studies have shown they are a major cause of febrile illness in West Africa 1.

1. Reddy EA, et al. Lancet Infect Dis 2010; 10: 417

We agree that testing only plasma samples limits us to detecting pathogens with a higher titer in the plasma. We selected plasma metagenomic sequencing for two reasons:

- We wanted to sample the blood, as many of the pathogens known to cause ambulatory febrile illness in Senegal, such as *Plasmodium* spp., Dengue Virus and relapsing fever *Borrelia*, reach high titers in the blood.
- We choose a cell-free fluid, because it has less host background and therefore enables more sensitive pathogen detection (Gu, Miller and Chiu, 2019).

We acknowledge that using plasma (a cell-free fluid) may limit our ability to detect intracellular pathogens. We have now modified our text to address this limitation in both the introduction and the discussion:

- Line 33-38: “As mNGS sequences all RNA or DNA in a sample, these techniques are typically less sensitive for detecting any single pathogen than targeted approaches (e.g., PCR), due to the abundance of the host relative to pathogen nucleic acids (Houldcroft, Beale and Breuer, 2017; Gu, Miller and Chiu, 2019). In order to achieve higher sensitivity and reduce the cost of deep sequencing, mNGS can be applied to cell-free fluids, which have lower host background, but this can limit the detection of intracellular pathogens.”
- Line 302-305: “Given that only cell-free plasma was sequenced, our approach may have missed infections limited to other body compartments, including the respiratory tract and digestive tract, and may have limited our ability to detect intracellular pathogens.”

Although the sensitivity to intracellular pathogens may be reduced in cell-free fluid, we are still able to detect intracellular bacteria such as *Rickettsia* in our study samples. A similar unbiased RNA-mNGS study on sera from febrile patients in Cambodia (Bohl *et al.*, 2022) detected intracellular bacteria, including *Salmonella*, *Orientia*, and *Rickettsia*. Notably, they only detected *Salmonella* in hospitalized patients and not in ambulatory patients. Although the absence of *Salmonella* in our study population is notable given the high burden in Sub-Saharan Africa, we believe our results are consistent with the lower incidence of bacteremia in general and *Salmonella* bacteremia specifically in ambulatory patients. We have now added a discussion of these points to the text (line 305-311):

“Though we did detect some intracellular pathogens (e.g., *Rickettsia*), we did not observe other intracellular bacterial pathogens, particularly *Salmonella* spp., which are a common cause of febrile illness in hospitalized patients in Sub-Saharan Africa (Reddy, Shaw and Crump, 2010;

Maze et al., 2018; Elven et al., 2020). Studies from elsewhere in Sub-Saharan Africa suggest a lower incidence of bacteremia in outpatients consistent with our study (D'Acremont et al., 2014; Mahende et al., 2015; Hildenwall et al., 2016; Maze et al., 2018), however, we cannot rule out that other intracellular bacterial infections were present at low titers in the blood and therefore not detected by plasma 16S and RNA-mNGS.”

The most commonly identified infections were *Borrelia*, which was found in 18/204 febrile patients (9%) and Rickettsial infections, both of which have been previously shown to be common causes of fever in Senegal.

While we agree that our finding of *Borrelia* in our study population is not novel, our study provides a uniquely in depth characterization of *Borrelia* infection, from epidemiology, to genome sequence, to immune response. Specifically:

- Previous studies of *Borrelia* in Senegal have examined either on *Borrelia* alone (Vial et al., 2006; Mediannikov et al., 2014; Ndiaye et al., 2021) or on a small subset of febrile pathogens (Sokhna et al., 2013; Abat et al., 2016).
- Our study is the first unbiased study in the primary care setting of febrile illness in Senegal, allowing us to characterize *Borrelia* in the context of other pathogens. By using unbiased RNA-metagenomics, targeted 16S metagenomics, and pan-*Borrelia* qPCR, we were able to characterize the prevalence of *Borrelia* alongside other blood borne pathogens, including bacteria, viruses, parasites, and co-infections.
- To our knowledge, our study is the first investigation of *Borrelia* in Thiès and, unlike prior studies, shows consistent prevalence of *Borrelia* across seasons and years.
- Our study also provides key insights on the sensitivity of metagenomic sequencing for detecting *Borrelia*, which will be increasingly important as metagenomics for pathogen surveillance becomes more widespread.
- Our work has the additional strength of collecting rich clinical information, cell counts, and immune chemokine/cytokine measurements across different pathogen types.
- Because this information was collected systematically over both seasons and two years, we were able to build a strong predictive model for *Borrelia* infection.
- This model highlights the key features distinguishing *Borrelia* from other non-malarial febrile illnesses and also demonstrates the potential for even better, clinically actionable models as we continue to collect more data in future studies.
- The information we provide here can help lay the foundations for better diagnostics to capture infections and study clinical outcomes of disease.
- Further, as the US and Europe are beginning to recognize the extraordinary impact and the long-term sequelae of *Borrelia* infection on patients, it is striking to find that approximately 8% of patients coming into primary care in Senegal suffer infection from the same pathogen family.

We thank the reviewer for helping us see that we did not clearly distinguish our results from the previous state of knowledge on *Borrelia* in Senegal. We have modified the introduction to emphasize the unique aspects of our work (line 49-52).

“Community and clinic based studies of patients with fever have revealed bacterial zoonoses are a common cause of ambulatory febrile illness in Senegal, including tick borne relapsing fever, Rickettsioses, Q fever, and Bartonella (Vial et al., 2006; Socolovschi et al., 2010; Sokhna et al., 2013; Mediannikov et al., 2014; Abat et al., 2016; Ndiaye et al., 2021). However, no unbiased surveillance of outpatient febrile illness has been done in Senegal to date.”

Microscopy of blood film, the current gold standard for the diagnosis of *Borrelia* infection, was shown to lack sensitivity compared to molecular methods but, since these are not widely available, the authors developed a weighted logistic regression model that considered 46 candidate predictors based on clinical symptoms, vital signs, and key demographic information to predict *Borrelia* infection, but it seems unlikely that this will be widely used at the primary care level, where a trial of treatment with tetracycline would be easier to deliver in a case of suspected borreliosis.

We thank the reviewer for pointing out that our presentation of the model directly following the discussion on sensitivity of smear, qPCR, 16S and mNGS may have implied our model should be used as an alternative diagnostic. We also agree that the model presented in our work is not currently ready for implementation in clinical practice. Rather, we intend this model to highlight the following points which we feel could be more immediately useful:

- While *Borrelia* has a highly overlapping clinical profile with malaria and other causes of febrile illness, looking at combinations of signs and symptoms can still help differentiate similar clinical entities.
- The model emphasizes the importance of a small number of features, such as vomiting in differentiating causes of disease. While this is not a replacement for a diagnostic test, we could share these key features with local clinicians to educate them on features that should heighten their suspicion for *Borrelia* and motivate additional diagnostic testing, such as smear examination or ideally qPCR.
- We note that at the same time, our lab has been developing field deployable tools for frontline healthcare workers to use multimodal data (Colubri et al., 2016, 2019).
- We hope to create similar tools for *Borrelia*, but we first need to develop better diagnostics to catch cases and gather the necessary clinical data on a large scale to build reliable multimodal models.

In order to avoid confusion, we have made the following edits to our text:

- We removed the transition statement “*Given the limited sensitivity of available smear-based diagnostics compared to molecular methods, we sought to assess the clinical syndrome associated with qPCR-confirmed Borrelia infections, compared to RDT-confirmed malaria and non-Borrelia NMFI (“other febrile”), to guide differential diagnosis at the point of care*” and change it to “*We sought to assess the clinical syndrome associated with qPCR-confirmed Borrelia infections, compared to RDT-confirmed malaria and non-Borrelia NMFI (“other febrile”), to guide differential diagnosis at the point of care*” (line 221-223)

- We changed “*We find that combinations of clinical signs and symptoms can increase suspicion for Borrelia and support targeting clinical care.*” to “*We find that combinations of clinical signs and symptoms can increase suspicion for Borrelia and support targeting **diagnostic testing and clinical care***” to clarify that we still feel diagnostics are needed as the model on its own is not clinically actionable (lines 331-332).

Regarding tetracycline treatment, these are a powerful therapeutic tool for *Borrelia* but come with several risks that we believe our findings may help to mitigate. Over half of patients with tick-borne relapsing fever (TBRF) treated with a tetracycline are at risk for Jarisch-Herxheimer reaction (JHR), which can be fatal. Additionally, tetracycline resistance is a growing global problem and the WHO has expressed concern over the overuse of antibiotics in non-malaria febrile illness (NMFI) patients, many of whom may not have a bacterial infection, which can contribute both to increasing resistance and to dissatisfaction with the healthcare system. We hope that the findings of our model and our discussion of sensitive methods for *Borrelia* diagnosis can help to more appropriately target tetracycline treatments to avoid these negative impacts and improve clinical care.

We have addressed these concerns about blanket tetracycline treatment without appropriate point of care diagnostics in our discussion (line 332-339):

“Due to the lack of available point-of-care tests, many clinics in LMICs rely on malaria RDTs as the primary diagnostic for febrile illness and may give blanket antibiotic treatment to RDT-negative patients (Hopkins et al., 2017). Concerns have been raised about this practice (WHO informal consultation on fever management in peripheral health care settings: A global review of evidence and practice, no date), including driving antibiotic resistance and exposing patients to unnecessary risks or side effects. For example, tetracycline antibiotic treatment—the recommended therapeutic for Borrelia and Rickettsial infections—poses the risk of Jarisch-Herxheimer reaction (JHR), a severe inflammatory response to spirochete lysis that often requires close patient monitoring (Butler, 2017).”

Reviewer #2 (Remarks to the Author):

The manuscript “Improving diagnosis of non-malarial fevers in Senegal: Borrelia and the contribution of tick-borne bacteria” is a well written and informative description of identification and sequence analysis of non-malarial pathogenic organisms identified in human specimens. The authors describe findings from two sequence based approaches that highlight common causes of febrile illness that may not be tested for or standard diagnostic approaches may miss. The authors highlight findings of relapsing fever group borrelia as a significant cause of febrile illness in the subset of specimens tested. This study highlights how sequence based approaches to clinical identification of infections can inform patient care and differential diagnosis of patients seeking care for febrile illness with a complex genetic diversity of potential infecting pathogens. The manuscript reads well, and I only had a few minor points of clarification I would like to see the authors address.

We thank the reviewer for the clear summary of our work and its key takeaways, and for your positive feedback. We greatly appreciate the suggestions for improving our manuscript and describe how we addressed each below.

Lines 80-82: It is unclear here if co-infections were identified between bacterial and viral positive specimens. Were the virus positive specimens tested for co-infection by 16S? Can you make more clear if this was the case. Were no viral/bacterial infections identified? How was this tested specifically?

We thank the reviewer for pointing out the lack of clarity in this section. We did not detect any viral/*Plasmodium* or viral-bacterial co-infections. All of the co-infections identified in 2019 are shown in Figure 1E.

Of the 204 febrile patient samples collected in 2019, all 204 underwent RNA-mNGS and *P. falciparum* RDT. We did not detect any viral/*Plasmodium* co-infections among these samples. Of these 204 samples, 129 underwent 16S sequencing, allowing us to additionally detect bacterial infections with more sensitivity. We did not detect any viral/bacterial co-infections in these samples. We have now added a line in the text explaining there are no viral/bacterial or viral/plasmodium coinfections (line 85-86).

“We did not detect any viral/bacterial or viral/Plasmodium coinfections.”

As we only performed 16S sequencing on samples with a high plasma bacterial load (measured with a qPCR targeting the v1-2 region of the 16S rRNA), 75/204 samples had only RNA-mNGS. It is possible there were viral/bacterial co-infections among these 75 samples that were missed by our sampling strategy. The approach to selecting samples for 16S sequencing is described in the methods (line 500-501):

“Samples with high bacterial load (V1-2 qPCR CT < 31.5) were selected for 16S sequencing (2019: febrile n = 129, healthy n = 35, 2018: febrile n = 84).”

Supplementary figure 7A is supposed to show the placement of the 16S V1V2 sequences from *Borrelia* positive specimens amongst other relevant *borrelia* species. The tree includes numerous species from the outlying “Lyme Disease” grouping. The majority of the species listed have never been shown to cause Lyme disease and, as labeled, the figure is very misleading. The figure should include minimal “Lyme group” species as it is an outgroup, and many more of the ~20 relapsing fever *borrelia* species. This would help the reader understand how 16S could potentially differentiate from other clinically relevant relapsing fever *borrelia* as V1V2 16S sequence should be sufficient enough to differentiate *B. crocidurae* from most others. This placement should then be compared to the confirmatory IGS, 23S sequencing presented in Fig 2C that include *B. duttonii* and *B. recurrentis*.

We thank the reviewer for pointing out the underrepresentation of the diversity of relapsing fever species in the tree. We initially used only sequences from the curated NCBI 16S rRNA dataset, but unfortunately relapsing fever species were poorly represented in this dataset.

We have now modified the tree to include one 16S v1-2 rRNA sequence from each specific relapsing fever species (ie not *Candidatus Borrelia* or *Borrelia* spp.) available in either (1) the NCBI 16S rRNA BioProject or (2) SilvaRefNR database. Instead of including numerous *Borrelia* species, we have instead included a single *Borrelia burgdorferi* sequence as an outgroup. We have also added additional annotations to the tree, including:

- Stars indicating which samples also underwent IGS sequencing so they can be easily compared
- Colors distinguishing soft-tick borne relapsing fever, hard-tick borne relapsing fever, reptile associated relapsing fever, and other zoonotic relapsing fever species.
- Text indicating samples with co-infections and the coinfecting taxa

We also made similar changes to the v1-2 16S rRNA phylogeny for *Rickettsia* in response to this comment and a similar comment from Reviewer #3 (now Supplementary Figure 7C). We feel these changes make both phylogenies much clearer and easier to interpret and thank the reviewer for the suggestion.

Supplementary Figure 7A: Maximum likelihood phylogenetic tree (IQ-TREE) of 16S v1-2 rRNA gene sequences for *Borrelia* from this study (febrile: black, healthy: grey) in the context of relapsing fever *Borrelia* sequences from the curated NCBI 16S rRNA target loci project and Silva non-redundant reference (Silva Ref NR) small subunit database (1 sequence per species) rooted on the outgroup, *Borrelia burgdorferi*. Stars indicate samples that also underwent IGS sequencing. In the case of multiple v1-2 sequences for a given taxa in a given sample, sequences are distinguished by “_1” and “_2”. In the case of co-infections, italic text next to the sample name lists the co-infecting taxa.

Upon viewing this improved v1-2 16S rRNA phylogeny, we noticed that a subset of samples had highly similar sequences to the sequence for *B. persica*. *B. persica* is a relapsing fever species from the Middle East, and would not be expected in Senegal. One of these samples, SHC0038, also underwent IGS sequencing. To evaluate whether SHC0038 is more similar to *B. crocidurae* or *B. persica*, we have added a modified IGS phylogeny with available IGS sequences for *B. persica* (Supplementary Figure 7B).

This tree shows that SHC0038 is indeed *B. crocidurae*, not *B. persica* as suggested by the v1-2 16S phylogeny. IGS phylogeny has been shown to discriminate between relapsing fever *Borrelia* species better than 16S phylogeny (Bunikis *et al.*, 2004), so we have concluded that SHC0038 is most likely *B. crocidurae*.

Supplementary Figure 7B: Maximum likelihood phylogenetic tree (IQ-TREE) IGS sequences from this study (black) in the context of available reference sequences for *B. crocidurae* (purple), *B. duttonii* (cyan), *B. recurrentis* (green), and *B. persica* (red), midpoint rooted.

We have modified the text (bold below) to clarify the comparison between the 16S and IGS phylogenies for *Borrelia* (line 182-185).

“To validate the 16S v1-2 phylogeny, a subset of 6 *Borrelia* positive samples were typed by amplicon sequencing of the 16S-23S intergenic spacer (IGS) (Bunikis et al., 2004). All sample sequences were *Borrelia crocidurae*, in agreement with the 16S v1-2 phylogeny (Fig. 2C, Supplementary Fig. 7B).”

Lines 179-183: The term “isolate” is used throughout this paragraph. Suggest replacing this term with a more accurate description used for culture free approach employed here. Unless cultured organisms were recovered for these six specimens, “isolate” seems inaccurate.

We have replaced “isolate” with “sample sequence” to accurately reflect the culture free approach.

Lines 205-212: Can the authors comment on whether qPCR was more sensitive than 16S sequencing based on these results in their hands? Did the additional 5 borrelia positive specimens that did not undergo 16S sequencing also not have RNA-mNGS data showing bacterial infection? I was unable to find supplementary figure 7E mentioned in the text for these specimens.

In order to address the question of relative sensitivity, we have added contingency tables for 16S sequencing (Supplementary Figure 6h) and RNA-mNGS (Supplementary Figure 6i) compared to pan-*Borrelia* qPCR. We also added a comment on the relative sensitivity of RNA-mNGS in 16S positive and qPCR positive samples (lines 191-193).

“*Borrelia* RNA-mNGS reads were detected in 68% (13/19) of 16S positive samples and 100% (17/17) of pan-*Borrelia* qPCR positive samples (Supplementary Fig 6i).”

		Borrelia 16S v1-2 rRNA abundance		Sensitivity: 0.82 Specificity: 0.97 PPV: 0.82 NPV: 0.97			Borrelia RNA-mNGS abundance		Sensitivity: 1.0 Specificity: 0.98 PPV: 0.71 NPV: 1.0
		≥ 5% (+)	< 5% (-)				≥ 1 rpm (+)	< 1 rpm (-)	
pan- Borrelia qPCR	Positive	28	6		pan- Borrelia qPCR	Positive	17	0	
	Negative	6	208			Negative	7	283	

Supplementary Figure 6: h. sensitivity and specificity of 16S sequencing and i. RNA-mNGS sequencing, compared to pan-*Borrelia* qPCR.

Of the five samples that were qPCR negative but did not undergo 16S sequencing, two samples underwent RNA-mNGS. Both samples had a high abundance of *Borrelia* RNA-mNGS reads (3243 rpm, 12284 rpm).

We also thank the reviewer for pointing out the mistaken reference to figure 7E and apologize for the error, we have removed this reference.

Lines 328-329: If symptomology and blood marker based differential criteria are not clinically actionable could this information still be disseminated for resource limited point-of-care clinicians to avoid potential JHR in RDT negative patients given RF borrelia were so frequently identified? This last sentence suggests the real issue is needing better point of care diagnostics for RF-borrelia. Does the symptomology differential lead to better test ordering?

We appreciate the reviewer's suggestions on how our findings could be applied to improve treatment. As noted in our response to a similar comment from Reviewer #1, we would like to clarify that we do not intend our model to be implemented as a clinical tool at this stage. With that in mind, we do hope that our data on blood markers and clinical symptoms can be read by clinicians in TBRF-endemic areas and help heighten their suspicions for *Borrelia* in certain patients. This could lead physicians to re-examine a smear or order a more sensitive diagnostic test. Ideally, we hope that point of care diagnostics and reliable qPCR assays will soon be available to clinicians working in these areas. We hope our findings can help motivate the development and deployment of new point-of-care diagnostic assays.

We appreciate that presenting the model directly after our discussion on the sensitivity of smear, qPCR, and next generation sequencing for detecting *Borrelia* may have implied we intended our model to be used as an alternative diagnostic. In order to avoid confusion, we have made the following edits to our text:

- We removed the transition statement "*Given the limited sensitivity of available smear-based diagnostics compared to molecular methods, we sought to assess the clinical syndrome associated with qPCR-confirmed Borrelia infections, compared to RDT-confirmed malaria and non-Borrelia NMFI ("other febrile"), to guide differential diagnosis at the point of care*" and made it "*We sought to assess the clinical syndrome associated with qPCR-confirmed Borrelia infections, compared to RDT-confirmed malaria and non-Borrelia NMFI ("other febrile"), to guide differential diagnosis at the point of care*" (line 221-223)
- We changed "*We find that combinations of clinical signs and symptoms can increase suspicion for Borrelia and support targeting clinical care.*" to "*We find that combinations of clinical signs and symptoms can increase suspicion for Borrelia and support targeting **diagnostic testing and clinical care***" to clarify that we still feel diagnostics are needed as the model on its own is not clinically actionable (lines 331-332).

Lines 330-338: Similar to the point above...The limitation of blood smears to differentiate borrelia species seems irrelevant to the point of this paragraph because RT-PCR does not necessarily provide species level resolution. The blood smear appeared to perform poorly compared to direct detection methods in this study and that seems more relevant. Genus specific RT-PCR seemed to perform quite well once an appropriate assay was used based on what the sequencing based approaches found. Please clarify the necessity of species level diagnostic resolution, if any, as it pertains to patient treatment and the feasibility of frontline diagnostic approaches and which ones would improve the current situation.

We agree with the reviewer that there are two distinct considerations regarding testing; sensitivity to detect *Borrelia* and ability to differentiate between species. Our study shows, as others have, that smear is very low sensitivity compared to nucleic acid diagnostics, such as qPCR. Further, our study is the first to our knowledge to show that unbiased RNA-mNGS is sensitive and specific for detection of relapsing fever *Borrelia*.

For clinical care, distinct *Borrelia* species are associated with different risks. For example, louse borne relapsing fever (LBRF) agent *B. recurrentis* has a higher mortality rate and different potential complications. Although *B. recurrentis* has not been identified in Senegal to date, the body louse vector exists wherever humans live. Further, work in Lyme Disease *Borrelia* has shown different clinical presentations and disease severity depending on the causative species. This may well be true for TBRF, but we will not be able to detect these differences and advise physicians accordingly unless species-specific diagnostics become routinely available. From a public health surveillance standpoint, knowing the currently circulating species in a given community is important to be able to detect events such as incursion of foreign *Borrelia* species on a new area or emergence of novel *Borrelia* species.

We had added a brief discussion of the potential importance of species-specific nucleic acid diagnostics for surveillance and clinical care (line 354-359).

“Louse-borne relapsing fever agent B. recurrentis (LBRF) has a higher fatality rate and risk of JHR than tick-borne relapsing fever (TBRF) species, but with smear alone clinicians cannot distinguish which disease they are treating unless the history of vector exposure is known. In Lyme Disease, molecular surveillance has shown that different species have different clinical presentations and severities (Radolf et al., 2021). Species-specific surveillance at scale could reveal analogous differences between relapsing fever Borrelia.”

Reviewer #3 (Remarks to the Author):

This interesting manuscript by Levine and colleagues reports on a RNA-metagenomic sequencing survey of blood samples of Senegalese febrile patients that were considered to not have malaria. The continued burden of fever in sites where antimalarial interventions are successful provide an excellent opportunity to discover other infections that might have been masked by the default malaria diagnosis; and, the use of modern diagnostic methods will potentially expand our knowledge of the association of other infectious agents with human disease.

We thank the reviewer for their interest in the work and the use of next generation sequencing for expanding our understanding of non-malarial febrile illness.

Although there is much emphasis on the confirmation of borreliosis as a cause of nonmalarial fevers, the finding is not new and indeed the literature is only selectively cited. Ndiaye et al 2021 reported 12% of 800 febrile samples contained evidence of a relapsing fever borrelia in another part of Senegal.

We thank the reviewer for pointing out our oversight of Ndiaye et al 2021 and agree that our finding of *Borrelia* in our study population itself is not novel. As noted in response to a similar comment from Reviewer #1, however, we feel our study provides a uniquely in depth characterization of *Borrelia* infection, from epidemiology, to genome sequence, to immune response. Specifically:

- Previous studies of *Borrelia* in Senegal have examined either on *Borrelia* alone (Vial *et al.*, 2006; Mediannikov *et al.*, 2014; Ndiaye *et al.*, 2021) or on a small subset of febrile pathogens (Sokhna *et al.*, 2013; Abat *et al.*, 2016).
- Our study is the first unbiased study in the primary care setting of febrile illness in Senegal, allowing us to characterize *Borrelia* in the context of other pathogens. By using unbiased RNA-metagenomics, targeted 16S metagenomics, and pan-*Borrelia* qPCR, we were able to characterize the prevalence of *Borrelia* alongside other blood borne pathogens, including bacteria, viruses, parasites, and co-infections.
- To our knowledge, our study is the first investigation of *Borrelia* in Thiès, and unlike prior studies, shows consistent prevalence of *Borrelia* across seasons and years.
- Our study also provides key insights on the sensitivity of metagenomic sequencing for detecting *Borrelia*, which will be increasingly important as metagenomics for pathogen surveillance becomes more widespread.
- Our work has the additional strength of collecting rich clinical information, cell counts, and immune chemokine/cytokine measurements across different pathogen types.
- Because this information was collected systematically over both seasons and two years, we were able to build a strong predictive model for *Borrelia* infection.
- This model highlights the key features distinguishing *Borrelia* from other non-malarial febrile illnesses and also demonstrates the potential for even better, clinically actionable models as we continue to collect more data in future studies.

- The information we provide here can help lay the foundations for better diagnostics to capture infections and study clinical outcomes of disease.
- Further, as the US and Europe are beginning to recognize the extraordinary impact and the long-term sequelae of *Borrelia* infection on patients, it is striking to find that approximately 8% of patients coming into primary care in Senegal suffer infection from the same pathogen family.

We have modified our introduction to emphasize the state of knowledge on *Borrelia* in Senegal and have cited these works appropriately (line 49-52).

“Community and clinic based studies of patients with fever have revealed bacterial zoonoses are a common cause of ambulatory febrile illness in Senegal, including tick borne relapsing fever, Rickettsioses, Q fever, and Bartonella (Vial et al., 2006; Socolovschi et al., 2010; Sokhna et al., 2013; Mediannikov et al., 2014; Abat et al., 2016; Ndiaye et al., 2021). However, no unbiased surveillance of outpatient febrile illness has been done in Senegal to date.”

We have added the Ndiaye et al 2021 citation (line 51, 178), as well as an additional citation of Abat et al. 2016. We hope these changes give the reader context to interpret our findings and the unique insights our study provides on *Borrelia* infection.

Similarly, although rickettsiosis is mentioned in multiple places in the manuscript as potentially burdensome (e.g., lines 164, 275, 281 et seq), there is again selective citation; Socolovskii et al. 2010 would be relevant for another area of Senegal, in which 8/134 nonmalarial febrile samples were considered to have evidence of infection by *R. felis*.

We cited Socolovschi et al. 2010 in our discussion of spotted fever *Rickettsia* in line 175 of our original submission (now line 180-181):

“Spotted fever Rickettsia, including R. felis (Socolovschi et al., 2010) and R. africae (Sokhna et al., 2013) have been previously detected in febrile Senegalese patients.”

We have added this citation to the modified introduction as well (line 51, see previous response).

(Of course, there is controversy regarding whether *R. felis* is actually a human pathogen, see Labruna and Walker 2014; this controversy should be discussed.)

We thank the reviewers for highlighting the controversy over *R. felis* as a human pathogen. This controversy is relevant to our finding that *Rickettsia* occurs more often as a co-infection than a standalone infection. We previously commented on the possibility that *Rickettsia* is simply a co-transmission of unknown pathogenic potential in our original submission (line 288) and have included additional text and citation of Labruna and Walker’s commentary in our revised submission (line 291-294).

“Alternatively, Labruna and Walker proposed that R. felis may not be a pathogen in and of itself, but rather a symbiont of a human parasite (Labruna and Walker 2014). Whether Rickettsia is contributing to pathogenesis or simply co-transmitting in these cases remains an open question.”

The phylogeny in supplemental figure 7B requires the inclusion of *R. felis*; it is not clear why why the authors chose not to include such a common and well known *Rickettsia* sp in an alignment and analysis of rickettsial sequences.

We thank the reviewer for pointing out the lack of *R. felis* in the tree and apologize for the oversight. We initially used only sequences from the curated NCBI 16S rRNA dataset, but this dataset did not include *R. felis*. We have modified the tree to include one 16S v1-2 rRNA sequence from each *Rickettsia* species associated with human disease (based on Abdad et al’s review (Abdad *et al.*, 2018)) available in either (1) the NCBI 16S rRNA BioProject or (2) SilvaRefNR database (Supplementary Figure 7C, previously 7B). Additionally, we have rooted the tree on *Orientia tsutsugamushi* as an outgroup. We have also added additional annotations to the tree, including:

- Color indicating group
- Text indicating which samples were co-infections and what the co-infecting taxa wer

Supplementary Figure 7C: Maximum likelihood phylogenetic tree (IQ-TREE) of 16S v1-2 rRNA gene sequences for *Rickettsia* from this study (febrile: black, healthy: grey) in the context of *Rickettsia* sequences from the curated NCBI 16S rRNA target loci project Silva Ref NR small subunit database (1 sequence per species, species associated with human disease only) rooted on the outgroup, *Orientia tsutsugamushi*. In the case of co-infections, italic text next to the sample name lists the co-infecting taxa.

We feel these changes make phylogeny much clearer and easier to interpret and thank the reviewer for the suggestion. We also made similar changes to the *Borrelia* v1-2 16S rRNA phylogeny (Supplementary Figure 7A).

In addressing this comment, we realized that we had included some very divergent *Rickettsia* sequences. While these sequences matched most closely to *Rickettsia* by BLAST, they do not meet the 94.5% threshold for similarity in the 16S rRNA within a genus, as proposed by Yarza et al 2014. Further, these sequences matched to uncultured *Rickettsia* rather than well-characterized *Rickettsia* known to infect humans. Based on these criteria, we chose to remove these sequences from the tree and change our classification of these patient samples.

We have modified the methods to reflect this change (line 523-525):

“Results were filtered to include only ASVs that had identity \geq 94.5% to the closest hit, as this has been proposed as a rational cutoff for 16S rRNA diversity within genera (Yarza et al., 2014).”

It is noteworthy that these febrile samples were so “clean”. Indeed, only 1 HIV positive and 2 hepatitis B positives were found, perhaps testament to good educational outreach by local healthcare workers? It seems to me that this is the real story that needs to be emphasized, that state of the art methods (the caveats about sensitivity of mNGS for intracellular agents within blood samples notwithstanding; do note that SFG rickettsiae may be detected within circulating endothelial cells sloughing off of vasculitic lesions into the blood) reveal a lack of association between known or even putative infectious agents and nonmalarial fevers. The very conservative discussion about possible detection of commensals is very appropriate and highlights the challenge of this kind of work. Perhaps mNGS is not necessarily going to provide us with etiology for all fevers caused by infectious agents, and clearly will not be useful for noninfectious causes (malignancy, heatstroke, inflammatory conditions, among others). Some speculative discussion might be useful to inform the general reader of other potential causes of fever. Why do we usually just focus on infection as a cause of fever? This has implications for primary care and for understanding the global burden of disease.

We appreciate the reviewer’s thoughtful discussion of the limitations of metagenomic approaches. We agree with the limitations of mNGS from a sample type perspective (ie only sampling blood) and analysis perspective (ie the difficulty of distinguishing fungal pathogens) and note these in our discussion. We thank the reviewer for further pointing out we did not elaborate on non-infectious causes of fever. We have added a brief discussion of this important point to the text (line 312-316).

“Lastly, noncommunicable diseases, such as cancer or autoimmune disease, can also cause fever. In clinical samples, negative mNGS results are common, but it can be difficult to determine whether these represent false-negatives due to limited sensitivity or true negatives(Oguzie et al., 2023). As the burden of non-communicable disease rises in West Africa(El Bcheraoui et al., 2020), the approach to diagnosing febrile illness must consider non-infectious etiologies.”

More broadly, we have extended our discussion about other limitations of mNGS for detecting causes of fever. Incorporating feedback from another reviewer (reviewer #1), we have focused this on limitations for detecting intracellular bacteria beyond *Rickettsia*.

- Line 33-38: *“As mNGS sequences all RNA or DNA in a sample, these techniques are typically less sensitive for detecting any single pathogen than targeted approaches (e.g., PCR), due to the abundance of the host relative to pathogen nucleic acids (Houldcroft, Beale and Breuer, 2017; Gu, Miller and Chiu, 2019). In order to achieve higher sensitivity and reduce the cost of deep sequencing, mNGS can be applied to cell-free fluids, which have lower host background, but this can limit the detection of intracellular pathogens.”*

- Line 302-311: *“Given that only cell-free plasma was sequenced, our approach may have missed infections limited to other body compartments, including the respiratory tract and digestive tract, and may have limited our ability to detect intracellular pathogens. Though we did detect some intracellular pathogens (e.g., Rickettsia), we did not observe other intracellular bacterial pathogens, particularly Salmonella spp., which are a common cause of febrile illness in hospitalized patients in Sub-Saharan Africa (Reddy, Shaw and Crump, 2010; Maze et al., 2018; Elven et al., 2020). Studies from elsewhere in Sub-Saharan Africa suggest a lower incidence of bacteremia in outpatients consistent with our study (D’Acremont et al., 2014; Mahende et al., 2015; Hildenwall et al., 2016; Maze et al., 2018), however, we cannot rule out that other intracellular bacterial infections were present at low titers in the blood and therefore not detected by plasma 16S and RNA-mNGS.”*

References

- Abat, C. *et al.* (2016) 'Implementation of Syndromic Surveillance Systems in Two Rural Villages in Senegal', *PLoS neglected tropical diseases*, 10(12), p. e0005212.
- Abdad, M.Y. *et al.* (2018) 'A Concise Review of the Epidemiology and Diagnostics of Rickettsioses: Rickettsia and Orientia spp', *Journal of clinical microbiology*, 56(8). Available at: <https://doi.org/10.1128/JCM.01728-17>.
- Bohl, J.A. *et al.* (2022) 'Discovering disease-causing pathogens in resource-scarce Southeast Asia using a global metagenomic pathogen monitoring system', *Proceedings of the National Academy of Sciences of the United States of America*, 119(11), p. e2115285119.
- Bunikis, J. *et al.* (2004) 'Typing of Borrelia relapsing fever group strains', *Emerging infectious diseases*, 10(9), pp. 1661–1664.
- Butler, T. (2017) 'The Jarisch-Herxheimer Reaction After Antibiotic Treatment of Spirochetal Infections: A Review of Recent Cases and Our Understanding of Pathogenesis', *The American journal of tropical medicine and hygiene*, 96(1), pp. 46–52.
- Colubri, A. *et al.* (2016) 'Transforming Clinical Data into Actionable Prognosis Models: Machine-Learning Framework and Field-Deployable App to Predict Outcome of Ebola Patients', *PLoS neglected tropical diseases*, 10(3), p. e0004549.
- Colubri, A. *et al.* (2019) 'Machine-learning Prognostic Models from the 2014-16 Ebola Outbreak: Data-harmonization Challenges, Validation Strategies, and mHealth Applications', *EClinicalMedicine*, 11, pp. 54–64.
- D'Acremont, V. *et al.* (2014) 'Beyond malaria--causes of fever in outpatient Tanzanian children', *The New England journal of medicine*, 370(9), pp. 809–817.
- El Bcheraoui, C. *et al.* (2020) 'Burden of disease in francophone Africa, 1990-2017: a systematic analysis for the Global Burden of Disease Study 2017', *The Lancet. Global health*, 8(3), pp. e341–e351.
- Elven, J. *et al.* (2020) 'Non-malarial febrile illness: a systematic review of published aetiological studies and case reports from Africa, 1980-2015', *BMC medicine*, 18(1), p. 279.
- Gu, W., Miller, S. and Chiu, C.Y. (2019) 'Clinical Metagenomic Next-Generation Sequencing for Pathogen Detection', *Annual review of pathology*, 14, pp. 319–338.
- Hildenwall, H. *et al.* (2016) 'Causes of non-malarial febrile illness in outpatients in Tanzania', *Tropical medicine & international health: TM & IH*, 21(1), pp. 149–156.
- Hopkins, H. *et al.* (2017) 'Impact of introduction of rapid diagnostic tests for malaria on antibiotic prescribing: analysis of observational and randomised studies in public and private healthcare settings', *BMJ*, 356, p. j1054.
- Houldcroft, C.J., Beale, M.A. and Breuer, J. (2017) 'Clinical and biological insights from viral genome sequencing', *Nature reviews. Microbiology*, 15(3), pp. 183–192.
- Mahende, C. *et al.* (2015) 'Bloodstream bacterial infection among outpatient children with acute febrile illness in north-eastern Tanzania', *BMC research notes*, 8, p. 289.

- Maze, M.J. *et al.* (2018) 'The epidemiology of febrile illness in sub-Saharan Africa: implications for diagnosis and management', *Clinical microbiology and infection: the official publication of the European Society of Clinical Microbiology and Infectious Diseases*, 24(8), pp. 808–814.
- Mediannikov, O. *et al.* (2014) 'Borrelia crocidurae infection in acutely febrile patients, Senegal', *Emerging infectious diseases*, 20(8), pp. 1335–1338.
- Ndiaye, E.H.I. *et al.* (2021) 'Tick-borne relapsing fever Borreliosis, a major public health problem overlooked in Senegal', *PLoS neglected tropical diseases*, 15(4), p. e0009184.
- Oguzie, J.U. *et al.* (2023) 'Metagenomic surveillance uncovers diverse and novel viral taxa in febrile patients from Nigeria', *Nature communications*, 14(1), p. 4693.
- Radolf, J.D. *et al.* (2021) 'Lyme Disease in Humans', *Current issues in molecular biology*, 42, pp. 333–384.
- Reddy, E.A., Shaw, A.V. and Crump, J.A. (2010) 'Community-acquired bloodstream infections in Africa: a systematic review and meta-analysis', *The Lancet infectious diseases*, 10(6), pp. 417–432.
- Socolovschi, C. *et al.* (2010) 'Rickettsia felis-associated unruptive fever, Senegal', *Emerging infectious diseases*, 16(7), pp. 1140–1142.
- Sokhna, C. *et al.* (2013) 'Point-of-care laboratory of pathogen diagnosis in rural Senegal', *PLoS neglected tropical diseases*, 7(1), p. e1999.
- Vial, L. *et al.* (2006) 'Incidence of tick-borne relapsing fever in west Africa: longitudinal study', *The Lancet*, 368(9529), pp. 37–43.
- WHO informal consultation on fever management in peripheral health care settings: A global review of evidence and practice* (no date). Available at: <https://www.who.int/publications-detail-redirect/9789241506489> (Accessed: 23 June 2023).
- Yarza, P. *et al.* (2014) 'Uniting the classification of cultured and uncultured bacteria and archaea using 16S rRNA gene sequences', *Nature reviews. Microbiology*, 12(9), pp. 635–645.
- Labruna, Marcelo B., and David H. Walker. 2014. "Rickettsia Felis and Changing Paradigms about Pathogenic Rickettsiae." *Emerging Infectious Diseases* 20 (10): 1768–69.

REVIEWERS' COMMENTS

Reviewer #1 (Remarks to the Author):

The authors responses to the reviewers' comments are clear and address all the issues raised in a satisfactory manner.

Reviewer #2 (Remarks to the Author):

All of my comments have been addressed in the re-submission. I have no further questions.

Reviewer #3 (Remarks to the Author):

The manuscript has been incrementally improved. The authors continue to hide the existing literature that has already made the point that relapsing fever is a main cause of non-malaria febrile illness in West Africa. A statement such as "we confirm that relapsing fever is a main cause of febrile illness in Senegal, as has been previously demonstrated" would be appropriate.

Other comments:

1. L279. Another main cause of NMFI in the study is stated to be *Rickettsia* spp. The authors then go on to discuss the possibility that *Rickettsia felis* may not be a pathogen. Is it not possible that the logistic modelling approach such as that done for borreliosis might help answer that question and would this not be useful for the paper?
2. L294-295. Not clear. "between the parasite and the rickettsia"...what parasite?
3. L295 et seq. In the context of this paragraph, the use of tetracyclines is entirely appropriate, it is the treatment of choice for borreliosis as well as rickettsiosis.
4. L334. Not clear. Are not the tetracyclines the treatment of choice for borreliosis, regardless of the risk of JHR?
5. L365 et seq. A bit speculative and grandiose. There is no evidence that the complex array of agents that may contribute to NMFI has changed over the years, we just have greater capacity to detect and definitively identify them. The old tropical medicine textbooks were very clear about the polymicrobial nature for etiology of the febrile patient.

Response to Reviewers' Comments

Reviewer #1 (Remarks to the Author):

The authors responses to the reviewers' comments are clear and address all the issues raised in a satisfactory manner.

We thank the reviewer for their time and helpful feedback.

Reviewer #2 (Remarks to the Author):

All of my comments have been addressed in the re-submission. I have no further questions.

We thank the reviewer for their time and helpful feedback.

Reviewer #3 (Remarks to the Author):

The manuscript has been incrementally improved. The authors continue to hide the existing literature that has already made the point that relapsing fever is a main cause of non-malaria febrile illness in West Africa. A statement such as "we confirm that relapsing fever is a main cause of febrile illness in Senegal, as has been previously demonstrated" would be appropriate.

We apologize that our previous revisions did not adequately address this issue, and we are very happy to make this more clear in this revision. We have now modified our statement in the discussion to make clear that our finding of a high proportion of relapsing fever *Borrelia* in febrile patients is not novel (line 288-289):

"Our results confirm that arthropod-borne bacterial pathogens, particularly relapsing fever *Borrelia*, are the major identifiable causes of NMFI, but remain underdiagnosed."

We have also included confirm in the title of the results subsection discussion our findings on *Borrelia* (line 172-173):

"16S sequencing confirms a high burden of Relapsing Fever *Borrelia* and Spotted Fever *Rickettsia*"

We have cited existing literature on relapsing fever *Borrelia* from Senegal specifically (Vial *et al.*, 2006; Sokhna *et al.*, 2013; Sambou *et al.*, 2014; Abat *et al.*, 2016; Ndiaye *et al.*, 2021) as well as literature from nearby countries with published genomic information (Bouattour *et al.*, 2010; Diatta *et al.*, 2012; Schwan *et al.*, 2012). We have also cited the reviews by Jakab *et al.* 2022

and Trape et al. 2013 on the epidemiology and clinical features of relapsing fever *Borrelia* (Trape et al., 2013; Jakab et al., 2022).

We thank the reviewer for their earlier suggestion to add Ndiaye *et al.* 2021. If there are additional studies the reviewer feels should be included we would welcome further feedback.

Other comments:

1. L279. Another main cause of NMFI in the study is stated to be *Rickettsia* spp. The authors then go on to discuss the possibility that *Rickettsia felis* may not be a pathogen. Is it not possible that the logistic modelling approach such as that done for borreliosis might help answer that question and would this not be useful for the paper?

We appreciate the reviewer’s suggestion, however unfortunately the number of *Rickettsia* infections is not sufficient for reliable logistic modeling. Instead we have performed a simple statistical test (fisher exact test) of the number of samples with a pathogen detected in febrile cases versus healthy controls. We have performed the same comparison between febrile cases that were malaria RDT positive and malaria RDT negative. We have included these results in an additional table, Supplementary Table 2:

Supplementary Table 2: Bacterial pathogens in febrile vs healthy patients and *P. falciparum* RDT positives vs RDT negative patients

Pathogen	Detection Method	Febrile vs healthy			P. falciparum RDT+ vs RDT- (febrile patients only)		
		Febrile samples positive	Healthy samples positive	p-value (Fisher exact, two-sided)	RDT+ samples positive	RDT- samples positive	p-value (Fisher exact, two-sided)
Borrelia	qPCR	7.4% (39/526)	0% (0/104)	0.0012	0% (0/75)	8.7% (39/448)	0.0033
Borrelia	16S sequencing	15.5% (33/213)	2.9% (1/35)	0.0593	16.7% (1/36)	18.2% (32/176)	0.0209
Rickettsia	16S sequencing	3.8% (8/213)	2.9% (1/35)	1.0000	5.6% (2/36)	3.4% (6/176)	0.6260

This analysis shows that there is a difference between *Borrelia* and *Rickettsia* in their distribution between cases and controls. Specifically, *Borrelia* (detected by both qPCR and 16S sequencing) was observed more often in febrile cases than healthy controls. This difference was statistically significant by pan-*Borrelia* qPCR, but borderline significant for 16S sequencing (likely due to the small number of controls that underwent 16S sequencing). We also see that *Borrelia* is significantly more common in RDT negative patients than RDT positive patients, suggesting it can cause pathogenesis on its own. Conversely, *Rickettsia* was not significantly more common in febrile cases than healthy controls and was not detected significantly more often in RDT negative patients compared to RDT positive febrile patients, making it difficult to determine whether *Rickettsia* is a stand alone pathogen from the available data from this study population.

We thank the reviewer for their initial suggestion to cite Labruna and Walker and we hope this analysis adds more clarity to the role of *Rickettsia* in our study population. Future studies will be needed to fully understand the pathogenic potential of *R. felis* and the interaction between *Plasmodium* and *R. felis* in human infections.

2. L294-295. Not clear. "between the parasite and the rickettsia"...what parasite?

We thank the reviewer for pointing out the nonspecific language. We have modified this section to more clearly communicate the hypothesis proposed by Labruna and Walker and the conclusions based on our data (line 300-305):

*"Alternatively, Labruna and Walker proposed that *R. felis* may not be a pathogen, but rather a symbiont of **parasites that infect humans, such as protozoa and helminths** (Labruna and Walker, 2014). Whether *Rickettsia* is contributing to pathogenesis or simply co-transmitting in the ***Rickettsia/Plasmodium* co-infections detected in this study** remains an open question. More research is needed to understand the interaction between **Plasmodium and *R. felis***."*

3. L295 et seq. In the context of this paragraph, the use of tetracyclines is entirely appropriate, it is the treatment of choice for borreliosis as well as rickettsiosis.

We thank the reviewer for pointing out this lack of clarity. We were referring specifically to the case of a *Rickettsia/Plasmodium* or *Borrelia/Plasmodium* co-infections where a patient may require either antimalarials, antibiotics, or both. We have modified the text to clarify this point (line 305-307):

*"Given the common occurrence of **bacterial/Plasmodium** co-infections and potential for negative outcomes without proper treatment of both pathogens, it is important that both surveillance and diagnostic approaches do not stop at detection of the first pathogen."*

4. L334. Not clear. Are not the tetracyclines the treatment of choice for borreliosis, regardless of the risk of JHR?

We thank the reviewer for this feedback. We realize that the order of our paragraph, specifically mentioning "unnecessary risks" directly before discussing the risk of JHR may have implied tetracyclines should be avoided in *Borrelia* positive patients because of the risk of JHR. We agree with the reviewer that tetracyclines are the treatment of choice, regardless of the risk of JHR.

Instead, we intended to make two separate statements. First, that access to diagnostics for *Borrelia* and other bacterial pathogens could reduce the unnecessary use of antibiotics in

patients who do not have a bacterial infection. Second, if a patient did test positive for *Borrelia*, a provider could give the appropriate tetracycline antibiotic and warn the patient about the risk of JHR.

We have modified the text to separate and clarify these statements (343-350):

*“Concerns have been raised about this practice(WHO informal consultation on fever management in peripheral health care settings: A global review of evidence and practice, no date), including driving antibiotic resistance and exposing patients **without evidence of bacterial infection** to unnecessary risks or side effects. **Even in patients with a bacterial infection**, tetracycline antibiotic treatment—the recommended therapeutic for *Borrelia* and Rickettsial infections—poses the risk of Jarisch-Herxheimer reaction (JHR), a severe inflammatory response to spirochete lysis(Butler, 2017). **Without accurate diagnosis of spirochetal infections, including *Borrelia*, patients and providers cannot anticipate and monitor for this complication.**”*

5. L365 et seq. A bit speculative and grandiose. There is no evidence that the complex array of agents that may contribute to NMFI has changed over the years, we just have greater capacity to detect and definitively identify them. The old tropical medicine textbooks were very clear about the polymicrobial nature for etiology of the febrile patient.

We thank the reviewer for pointing out that many of these pathogens have been circulating for decades, if not longer, and we completely agree. We intended to convey that as a result of the decreasing burden of malaria, the relative burden of different agents may be changing. Further, as the review noted, we now have more capacity to detect different agents and more treatment options available, meaning that detecting and distinguishing different etiologies of NMFI can result in better patient care and public health interventions.

We have removed the first line and altered the following text to more accurately convey this point (375-378):

*“Decreased malaria transmission coupled with changes in the range and incidence of other pathogens—influenced in part by global climatic changes—**may be shifting the burden of pathogens causing febrile disease**(Rocklöv and Dubrow, 2020). **New technologies have greatly increased our ability to detect both established and emerging pathogens.**”*

Other changes

During the course of the revision process, we were able to perform additional analyses to verify the *Plasmodium* RDT negative and RNA-mNGS positive (RDT-/RNA-mNGS+) samples. We have expanded this section of the results (line : 147-158), added additional panels to Supplementary Figure 5, and added Supplementary Table 3 to describe our findings:

“In order to determine why 11 samples were negative by RDT but positive by RNA-mNGS, we first confirmed the results with a pan-Plasmodium qPCR. We found that 8/11 samples were positive by pan-Plasmodium qPCR and the cycle threshold (CT) correlated with Plasmodium RNA-mNGS rpm (Supplementary Figure 5D). Next, we determined whether any samples belonged to species other than Plasmodium falciparum, the target species of the RDTs, using both RNA-mNGS reads and a P. falciparum-specific qPCR. As previous work has demonstrated Kraken2 is inaccurate for species-level classification of parasites (Frickmann et al., 2022), we used DIAMOND to identify one sample with a contig (771 bp) that matched perfectly (100% nucleotide identity, 100% coverage) to Plasmodium ovale cytochrome oxidase subunit 1 (Accession: KP050416.1), suggesting a likely P. ovale infection. This sample was positive on the pan-Plasmodium qPCR assay, but negative on the P. falciparum qPCR, confirming it as a case of non-falciparum malaria (Supplementary Table 3, Supplementary Fig 5E).”

Supplementary Table 3: Investigation of RDT negative Plasmodium infections

Sample	Case or control?	Age group	Season	Plasmodium RNA-mNGS reads	Prior RDT?	Number of previous RDTs	Prior antimalarials?	pan-Plasmodium qPCR result [#]	P. falciparum qPCR result [#]
SHC0755	Febrile	Adult	Dry	6.1E+02	No		No	FALSE	FALSE
SHC0758	Febrile	Young child	Dry	9.8E+02	No		No	FALSE	FALSE
SHC1007	Febrile	Child	Rainy	1.8E+03	No		No	TRUE	TRUE
SHC1009	Febrile	Adult	Rainy	1.6E+03	No		No	TRUE	TRUE
SHC1015	Febrile	Young child	Rainy	1.0E+03	No		No	TRUE	TRUE
SHC1016	Febrile	Adult	Rainy	2.6E+03	No		No	TRUE	TRUE
SHC1018	Febrile	Child	Rainy	2.1E+03	No		No	TRUE	TRUE
SHC1019	Febrile	Adolescent	Rainy	1.7E+03	No		missing	TRUE	TRUE
SHC1025	Febrile	Adult	Rainy	1.4E+04	No		No	TRUE	TRUE
SHC1056	Febrile	Adult	Rainy	1.7E+04	Yes	5	No	TRUE	FALSE
SHT1153	Healthy	Adult	Rainy	6.8E+02	missing		missing	FALSE	FALSE

*Young child: 2-6, child: 6-12 years, adolescent: 13-17 years, adult 18+

CT < 40 in 3/3 technical replicates is a positive test

Supplementary Figure 5: d. *Plasmodium* RNA-mNGS reads per million (plasma) vs pan-*Plasmodium* CT (dried blood spot) and **e.** pan-*Plasmodium* CT vs *P. falciparum* CT. Dashed lines represent the cutoffs for considering a sample positive. For qPCR assays, samples were considered positive if CT was < 40 in 3/3 technical replicates and mean CT across replicates is plotted for each sample. For negative samples, CT was set to 41 for visualization. Regressions were performed only on positive samples.

We additionally assessed two common deletion breakpoints in *pfhrp2* and a flanking gene, as previously described by Cheng et al. (Cheng et al., 2014). We have added a description of these results (line 165-170) and Supplementary Figure 5G:

“To assess for the presence of deletions at common breakpoints, we amplified two targets, one in exon 2 of *pfhrp2* gene and one in a flanking gene which can also be included in deletions (Cheng et al., 2014). Amplification of all both targets was intact in all qPCR-confirmed *P. falciparum* infections (7/7, Supplementary Figure 5G), suggesting none of these parasites had deletions in these targets consistent with the low frequency of *pfhrp2* deletion in Senegal, previously estimated at 2.4% (Agaba et al., 2019).

g

Supplementary Figure 5G: Amplification of *pfhrp2* exon 2 and the flanking gene Pf3D7_0831700, visualized on E-gel EX 2% with E-gel 1kb plus ladder. RDT+/mNGS+ samples and 3D7, a *P. falciparum* line with intact *pfhrp2*, are shown for comparison.

References:

- Abat, C. *et al.* (2016) 'Implementation of Syndromic Surveillance Systems in Two Rural Villages in Senegal', *PLoS neglected tropical diseases*, 10(12), p. e0005212.
- Agaba, B.B. *et al.* (2019) 'Systematic review of the status of pfhrp2 and pfhrp3 gene deletion, approaches and methods used for its estimation and reporting in Plasmodium falciparum populations in Africa: review of published studies 2010-2019', *Malaria journal*, 18(1), p. 355.
- Bouattour, A. *et al.* (2010) 'Borrelia crocidurae infection of Ornithodoros erraticus (Lucas, 1849) ticks in Tunisia', *Vector borne and zoonotic diseases*, 10(9), pp. 825–830.
- Butler, T. (2017) 'The Jarisch-Herxheimer Reaction After Antibiotic Treatment of Spirochetal Infections: A Review of Recent Cases and Our Understanding of Pathogenesis', *The American journal of tropical medicine and hygiene*, 96(1), pp. 46–52.
- Cheng, Q. *et al.* (2014) 'Plasmodium falciparum parasites lacking histidine-rich protein 2 and 3: a review and recommendations for accurate reporting', *Malaria journal*, 13, p. 283.
- Diatla, G. *et al.* (2012) 'Epidemiology of tick-borne borreliosis in Morocco', *PLoS neglected tropical diseases*, 6(9), p. e1810.
- Frickmann, H. *et al.* (2022) 'Metagenomic Sequencing for the Diagnosis of Plasmodium spp. with Different Levels of Parasitemia in EDTA Blood of Malaria Patients-A Proof-of-Principle Assessment', *International journal of molecular sciences*, 23(19). Available at: <https://doi.org/10.3390/ijms231911150>.
- Jakab, Á. *et al.* (2022) 'Tick borne relapsing fever - a systematic review and analysis of the literature', *PLoS neglected tropical diseases*, 16(2), p. e0010212.
- Labruna, M.B. and Walker, D.H. (2014) 'Rickettsia felis and changing paradigms about pathogenic rickettsiae', *Emerging infectious diseases*, 20(10), pp. 1768–1769.
- Ndiaye, E.H.I. *et al.* (2021) 'Tick-borne relapsing fever Borreliosis, a major public health problem overlooked in Senegal', *PLoS neglected tropical diseases*, 15(4), p. e0009184.
- Rocklöv, J. and Dubrow, R. (2020) 'Climate change: an enduring challenge for vector-borne disease prevention and control', *Nature immunology*, 21(5), pp. 479–483.
- Sambou, M. *et al.* (2014) 'Identification of rickettsial pathogens in ixodid ticks in northern Senegal', *Ticks and tick-borne diseases*, 5(5), pp. 552–556.
- Schwan, T.G. *et al.* (2012) 'Endemic foci of the tick-borne relapsing fever spirochete Borrelia crocidurae in Mali, West Africa, and the potential for human infection', *PLoS neglected tropical diseases*, 6(11), p. e1924.
- Sokhna, C. *et al.* (2013) 'Point-of-care laboratory of pathogen diagnosis in rural Senegal', *PLoS neglected tropical diseases*, 7(1), p. e1999.
- Trape, J.-F. *et al.* (2013) 'The epidemiology and geographic distribution of relapsing fever borreliosis in West and North Africa, with a review of the Ornithodoros erraticus complex (Acari: Ixodida)', *PloS one*, 8(11), p. e78473.

Vial, L. *et al.* (2006) 'Incidence of tick-borne relapsing fever in west Africa: longitudinal study', *The Lancet*, 368(9529), pp. 37–43.

WHO informal consultation on fever management in peripheral health care settings: A global review of evidence and practice (no date). Available at:
<https://www.who.int/publications-detail-redirect/9789241506489> (Accessed: 23 June 2023).